# Dynamical patterns and nonreciprocal effective interactions in an active-passive mixture through exact hydrodynamic analysis

James Mason[1], Robert L. Jack[1,2] & Maria Bruna [1,3] ✉

The formation of dynamical patterns is one of the most striking features of nonequilibrium physical systems. Recent work has shown that such patterns arise generically from forces that violate Newton's third law, known as nonreciprocal interactions. These nonequilibrium phenomena are challenging for modern theories. Here, we introduce a model mixture of active (self-propelled) and passive (diffusive) particles amenable to exact mathematical analysis. We exploit state-of-the-art methods to derive exact hydrodynamic equations for the particle densities, which reveal effective nonreciprocal couplings between the active and passive species. We study the resulting collective behavior, including the linear stability of homogeneous states and phase coexistence in large systems. This reveals a novel phase diagram with the spinodal associated with active phase separation protruding through the associated binodal, heralding the emergence of dynamical steady states. We analyze these states in the thermodynamic limit of large system size, showing, for example, that sharp interfaces may travel at finite velocities, but traveling phase-separated states are forbidden. The model's mathematical tractability enables precise new conclusions beyond those available by numerical simulation of particle models or field theories.

Simple systems of interacting particles (or agents) can support complex emergent behavior, including the self-assembly of nanoscale equilibrium structures, the self-organization of animals into flocks and swarms, and pattern formation in chemical reactions. Describing these effects has been a long-standing challenge for physics and mathematics: modern theories focus on emergent nonequilibrium behavior, making it difficult to predict macroscopic collective phenomena from the underlying microscopic rules. Recent studies have highlighted that nonreciprocal interactions in nonequilibrium systems lead generically to pattern formation in a variety of physical settings[1], including reaction-diffusion systems[2-4], living chiral crystals[5,6] and quorum sensing bacteria[7,8]. Despite their diversity, these systems appear to self-organize according to a common set of physical principles, offering the opportunity for a predictive theory with broad scope.

Within this context, the nonreciprocal Cahn–Hilliard (NRCH) equation has recently emerged as a canonical model for nonreciprocally coupled particle models[9-11]. It illustrates that phase-separated (PS) systems can undergo exceptional phase transitions when subjected to nonreciprocal driving, leading to dynamical (traveling) steady states. This simple and elegant equation bridges the established equilibrium theory of phase separation and the complex world of pattern formation in nonequilibrium systems. However, pattern formation in nonreciprocally coupled systems continues to

[1]DAMTP, Centre for Mathematical Sciences, University of Cambridge, Cambridge, UK. [2]Yusuf Hamied Department of Chemistry, University of Cambridge, Cambridge, UK. [3]Mathematical Institute, University of Oxford, Oxford, UK. ✉e-mail: bruna@maths.ox.ac.uk

challenge our understanding, including the vital question of which patterns will appear in any given system[1,12–14].

This work addresses these challenges by analyzing a specific nonreciprocal system for which exact mathematical results can be derived. Specifically, we introduce an idealized mixture of interacting active and passive particles and derive its exact hydrodynamic limit, the equation that governs its large-scale collective behavior. Such mixtures are known to display many features of nonreciprocally interacting systems, and NRCH-like equations that approximate their motion have been proposed, either on phenomenological grounds or by various approximation schemes[9,15–17]. These capture many qualitative features of the resulting dynamics but are only partially quantitative. Our hydrodynamic equation leads to phenomenology similar to that of NRCH and is consistent with generic principles of self-organization via nonreciprocity.

Our analysis relies on state-of-the-art mathematical techniques together with a simple underlying model. In particular, we assume that particles move on an underlying two-dimensional lattice with stochastic dynamical rules and that self-propulsion only occurs in the left and right directions. These idealized features enable a rigorous hydrodynamic limit[18,19]: when the lattice spacing tends to zero and the number of particles tends to infinity, the particle densities obey deterministic continuum equations, which we derive exactly. The resulting system differs from NRCH: it has some similar features but also reveals interesting new behavior, which we outline next.

In our model, active particles alone result in stationary, motility-induced phase separation (commonly referred to as motility-induced phase separation or MIPS[20]), but adding an extra population of passive (diffusing) particles can induce patterns, including traveling clusters, where self-organized groups of active particles push their passive counterparts around the system[21,22], see also ref. [23]. The hydrodynamic equation reveals an unusual type of phase diagram where the spinodal curve for the liquid-vapor transition protrudes through the binodal, signaling the onset of pattern formation. In this region, we demonstrate the existence of asymmetric traveling patterns and patterns formed of counter-propagating (CP) clusters. For large systems, these patterns feature sharp interfaces that separate dense liquid and dilute vapor regions. By studying the hydrodynamic equation in large domains, we relate traveling interfaces to those of static MIPS clusters, providing a new link between the equilibrium-like physics of phase separation and the dynamical patterns characteristic of nonreciprocity.

Our study provides an explicit and important example of how nonreciprocal effective interactions can emerge at the macroscale from simple interactions at the microscale. Specifically, while the direct interactions among particles are reciprocal, resulting solely from volume exclusion, particle simulations and analyses of the hydrodynamic equation clearly demonstrate that the model exhibits the principles of nonreciprocal self-organization. Furthermore, its idealized features allow us to draw sharp conclusions about large length and time scales, which would be extremely challenging to achieve from numerical simulations of more complex models. Our analysis of the large-system limit demonstrates how concepts from equilibrium phases and their interfaces can be applied to pattern-forming states. These features also facilitate a fully nonlinear treatment of the pattern-forming (traveling) steady states, revealing much more intricate behavior than could be predicted by linear stability analysis of the homogeneous state.

Based on these exact results, we identify several phenomena that should be considered as reference points for future studies of non-reciprocal systems. Specifically: the protrusion of a spinodal line through the binodal is a generic mechanism for instability of static phase separation, leading to dynamical patterns; in large systems, such patterns may incorporate interfaces between dense and dilute regions resembling those found in static phase separation; the resulting

patterns may travel with a fixed velocity or they may include different traveling objects that interact repeatedly. These results significantly expand the phenomenology of nonreciprocally coupled mixtures. The separation of length scales between sharp interfaces and macroscopic dense/dilute domains suggests focusing on dynamical patterns in a suitable large-system limit in order to draw useful analogies with static (thermodynamics) phase transitions. We discuss these points further in later sections.

## Results

### APLG model

We consider an active-passive lattice gas (APLG) model that extends the active lattice gases of refs. [19,24,25]. It is defined on a two-dimensional periodic square lattice with spacing $h$. Placing at most one particle per site, we populate the lattice with three types of particles $\sigma \in \{+1, -1, 0\}$: active particles oriented right ($\sigma = +1$), active particles oriented left ($\sigma = -1$) and passive particles ($\sigma = 0$). Each lattice site $(i, j)$ has a position $\mathbf{x} = (ih, jh) \in [0, \ell_x] \times [0, \ell_y]$. The model dynamics can be split into four parts: (i) passive particles attempt nearest neighbor jumps with jump rate $D_T/h^2$ per adjacent site, where $D_T$ is the spatial diffusion constant; (ii) active particles perform nearest neighbor random walk, weakly biased in the direction of their orientation to account for self-propulsion. In particular, a jump in the $\mathbf{u}$ direction (where $|\mathbf{u}| = h$) is attempted at rate $D_T/h^2 + \frac{v_0}{2h^2}(\mathbf{u} \cdot \mathbf{e}_\sigma)$, where $\mathbf{e}_\sigma = (\sigma, 0)$ is the particle's orientation, and $v_0$ is the self-propulsion speed; (iii) both types of particles are under an exclusion rule: if the target site of a jump is occupied, the jump is aborted; if the site is otherwise empty, the jump is executed; and (iv) each active-particle orientation flips at rate $D_R$. The total numbers of active and passive particles are specified via their volume fractions $\phi_a$ and $\phi_p$, respectively, and the overall volume fraction is $\phi = \phi_a + \phi_p \in [0, 1]$.

**Hydrodynamic limit.** The number of lattice sites in the APLG is $\ell_x \ell_y / h^2$, and we identify the lattice spacing $h$ with the size of a particle. The hydrodynamic limit is $h \to 0$ at fixed $\ell_x, \ell_y, \phi_a, \phi_p$: it corresponds to "zooming out", in order to describe motion on scales much larger than the particle size. The APLG dynamics above have nontrivial $h$-dependence [for example, the hopping rate is $O(h^{-2})$ but the orientation flip rate is $O(1)$]. This choice of scaling limit enables rigorous mathematical calculations[19]. Other types of large-system limit are discussed in Supplementary Information Sec. A1.

To take the hydrodynamic limit, it is convenient to rescale time and space by $D_R^{-1}$ and $\sqrt{D_T/D_R}$ respectively, and introduce the Péclet number $\text{Pe} = v_0/\sqrt{D_T D_R}$. A configuration of the APLG is defined in terms of occupancies: $\eta_\sigma(\mathbf{x}, t) \in \{0, 1\}$ is the number of particles of type $\sigma$ at site $\mathbf{x}$ and time $t$. The hydrodynamic limit equations describe the evolution of the local densities $\rho_\sigma(\mathbf{x}, t)$ of particles of type $\sigma \in \{+1, -1, 0\}$, as the lattice spacing $h \to 0$. Building on ref. [19], we rigorously derive the hydrodynamic limit of the APLG model, obtaining exact macroscopic evolution equations for the densities $\rho_\sigma$ (see Supplementary Information Sec. A)

$$\partial_t \rho_\sigma = \nabla \cdot \left[ d_s(\rho) \nabla \rho_\sigma + \rho_\sigma \mathcal{D}(\rho) \nabla \rho \right] - \text{Pe}\, \partial_x \left[ \rho_\sigma s(\rho) m + \sigma d_s(\rho) \rho_\sigma \right] - \sigma m, \tag{1}$$

with periodic boundary conditions. Here $\nabla = (\partial_x, \partial_y)$, $m = \rho_+ - \rho_-$ is the magnetization, we also define $\rho_a = \rho_+ + \rho_-$ as the active-particle density, so $\rho = \rho_a + \rho_0$ is the total particle density. Further, $d_s(\rho)$ is the self-diffusion coefficient of a simple symmetric exclusion process[19,26] and

$$\mathcal{D}(\rho) = [1 - d_s(\rho)]/\rho, \quad s(\rho) = \mathcal{D}(\rho) - 1. \tag{2}$$

Without activity (Pe = 0), the APLG model reduces to a three-species symmetric simple exclusion process, so it is natural that $d_s$ appears in

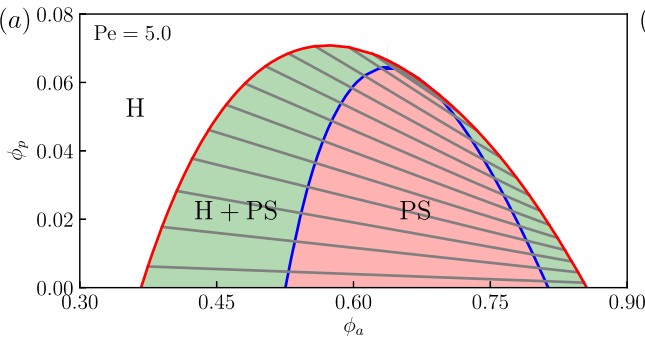

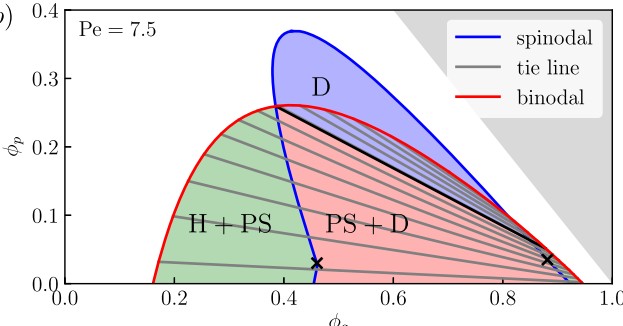

**Fig. 1 | Phase diagrams of the active-passive mixture for two values of Pe.** We show the diagram spanned by the active and passive particles' volume fractions $\phi_a$ and $\phi_p$, respectively, for (**a**) Pe = 5 and (**b**) Pe = 7.5. The spinodal (blue curve) encloses the region of linear stability of homogeneous solutions. The binodal (red curve) encloses the region of phase separation. Intersection points of the tie lines (gray lines) with the binodal indicate the composition of the liquid and vapor phases. Black crosses indicate bifurcations of co-dimension two; see text for

discussion. Regions with stable Homogeneous (H), Phase Separated (PS), and Dynamical (D) steady states are marked on the diagram. [In (**b**), the white region is left unlabeled because the boundary between H+D and H is unknown, and the gray region corresponds to $\phi_a + \phi_p > 1$, which is inaccessible due to size exclusion. The black tie line intersects the spinodal, so phase-separated states above this line are linearly unstable].

its hydrodynamic limit[26]. Technically, the APLG model is of non-gradient type in the sense of ref. [18]; this means that while (1) is exact, there is no explicit expression for $d_s(\rho)$, although variational formulae are available[27,28]. Even so, $d_s$ may be approximated to arbitrary accuracy, either numerically or analytically. We use the polynomial approximation[29]:

$$d_s(\rho) \approx (1-\rho)\left(1 - \alpha\rho + \frac{\alpha(2\alpha-1)}{2\alpha+1}\rho^2\right), \qquad (3)$$

with $\alpha = \pi/2 - 1$, which exactly matches the value and the first derivative of $d_s(\rho)$ at $\rho = 0, 1$, and approximates $d_s$ very accurately for all $\rho$. For other accurate approximations, see refs. [30–35]. Here, we use (3) in the numerical analysis of the exact hydrodynamic limit (1). Corrections to the mean-field approximation $\langle\eta_\sigma(\mathbf{x},t)\eta_\sigma(\hat{\mathbf{x}},t)\rangle \approx \rho_\sigma(\mathbf{x},t)\rho_\sigma(\hat{\mathbf{x}},t)$ are important in the APLG model and enter the hydrodynamic limit; the mean-field approximation would lead to $d_s(\rho) = (1-\rho)$. The method to obtain (1) is valid for any dimension greater than one.

Mathematical analysis of (1) is challenging due to the density dependence of the coefficients. However, since the active self-propulsion is only in the horizontal ($x$) direction, the equations are diffusive in the vertical direction[26], so that any variation with respect to $y$ will converge to zero over time, and instabilities of the homogeneous state are also independent of $y$. We exploit this symmetry throughout, restricting solutions of (1) to the form $\rho_\sigma(x,y,t) = \rho_\sigma(x,t)$. Four dimensionless parameters govern these solutions: the Péclet number Pe, the rescaled domain length in the horizontal axis $L = \ell_x\sqrt{D_R/D_T}$, the active volume fraction $\phi_a$, and the passive volume fraction $\phi_p$.

**Overview of phase behavior**

The APLG model supports different dynamical phases. We focus on behavior in the thermodynamic limit of large domains ($L \gg 1$), which allows some properties to be derived analytically. As in pure active systems[25], the APLG supports PS stationary states with macroscopic dense (liquid) and dilute (vapor) regions. The densities of active and passive particles in the liquid and vapor phases trace out the binodal curve in the ($\phi_a$, $\phi_p$) plane, see Fig. 1a. For $L \gg 1$, this curve can be calculated numerically exactly following the prescription of ref. [36]. This relies on the observation that, for PS states, local concentrations of passive particles and vacancies are proportional

$$\rho_0(x) = \nu[1 - \rho(x)] \qquad (4)$$

(see Methods), with

$$\nu = \phi_p/(1-\phi). \qquad (5)$$

We also compute the spinodal curve as the limit of stability of the homogeneous state (which is $\rho_\pm = \phi_a/2$, $\rho_0 = \phi_p$). Linear stability analysis of (1) shows that the homogeneous state is unstable when Pe is sufficiently large and $\phi_p$ is sufficiently small (see Methods). For the case shown in Fig. 1a, the dominant eigenvalue of the stability problem is always real, as in the pure active case. Then, the binodal and spinodal are tangent at the critical point (see Fig. 1a), which is the standard phenomenology of liquid-vapor phase coexistence for two-component mixtures. Inside the spinodal, the homogeneous state is linearly unstable, and the PS state is stable; between the spinodal and binodal, the PS state is globally stable while the homogeneous state is metastable.

For larger Pe, this familiar scenario changes qualitatively. In particular, the dominant eigenvalues of the linear stability problem may become complex, leading to dynamical steady states. Such a scenario is illustrated in Fig. 1b. Crosses on the spinodal mark co-dimension two (Bogdanov–Takens) bifurcations[37] where the dominant eigenvalue becomes complex and the instability of the homogeneous state changes from a pitchfork to a Hopf bifurcation[38]. Crucially, the binodal curve can still be computed for this system: we observe that the spinodal protrudes through the binodal. This effect is intrinsically linked to the existence of nonreciprocal effective interactions between active and passive densities and signals the onset of new physics[9,15,22]. (See Supplementary Information Sec. D for details of the stability analysis.)

Within the protruding part of the spinodal, the homogeneous state is linearly unstable, and PS states do not exist. It follows that some dynamical steady states must be present in this region. Moreover, in regions where PS states do exist, it may be that (at least) one of the coexisting phases is linearly unstable, which also renders the PS state unstable. The result is that dynamical steady states must exist throughout the blue-shaded region D in Fig. 1b. Note that this does not rule out the existence of dynamical states elsewhere. Also, while the dominant eigenvalue for the instability is complex for a large part of the spinodal in Fig. 1b, the resulting steady state may still be (stationary) PS, showing that steady state properties cannot be deduced directly from linear stability analysis. Dynamical behavior similar to the APLG also occurs in other nonreciprocal systems[9,15,22]: we will see below that the APLG provides a specific microscopic model where such

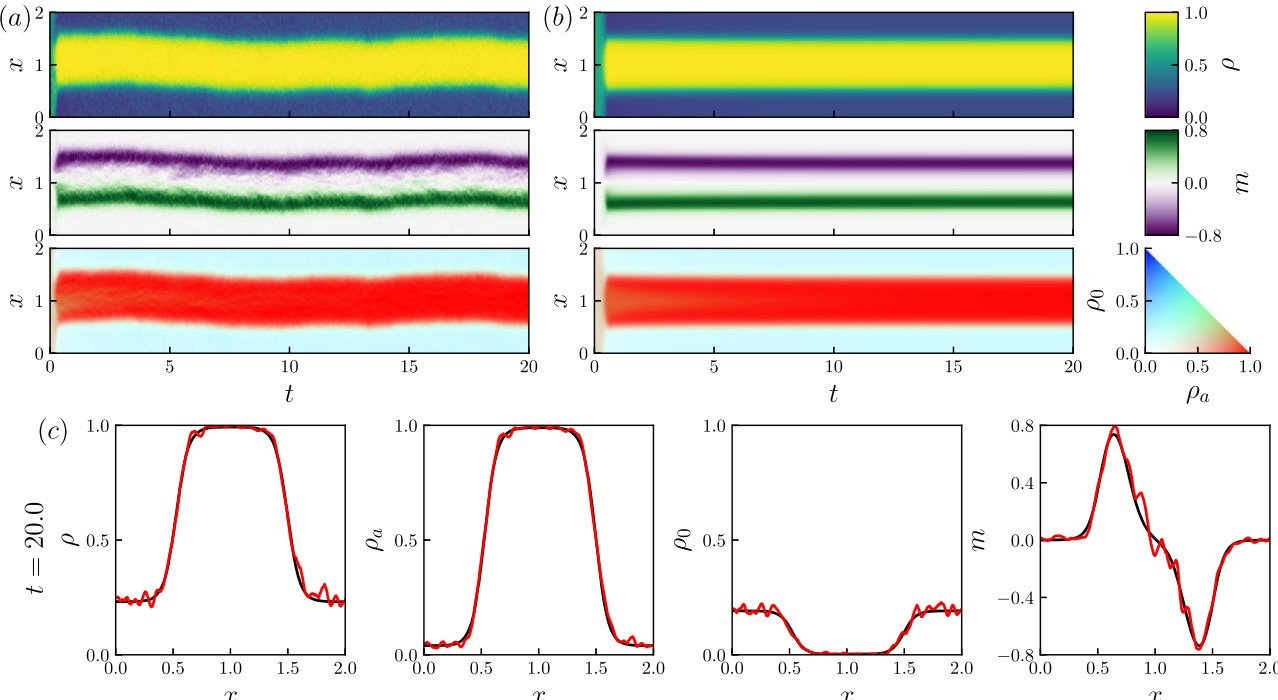

**Fig. 2 | Phase-separated stationary solution. a, b** Kymographs showing the spatiotemporal dynamics of a phase-separating solution. **a** $y$-averaged local density of a particle simulation (see Methods). **b** Numerical solution to (1), the initial condition is a homogeneous state with a uniform random perturbation (see Supplementary Information Sec. E1 for details, Eq. (S62)). **c** Density profile at $t = 20$. Black lines display the solution to (1), and red lines display the local density of the particle simulation. Parameters: Pe = 20, $L = 2$, $\phi_a = 0.5$, $\phi_p = 0.1$, $\Delta x = 0.01$ (PDE model), $h = 0.01$ (particle model).

generic phenomena can be analyzed quantitatively, including in the thermodynamic limit of large system size.

We emphasize that the spinodal marks the stability limit of homogeneous states, but PS stationary states can also exhibit other linear instabilities corresponding to critical exceptional points or exceptional phase transitions[1,9,39]. In such transitions, broken translational symmetry of the PS state plays an important role. We expect the (blue) region of purely dynamical steady states in Fig. 1b to extend to lower $\phi_p$. We discuss such cases below, as well as other state points where both static and dynamical states are linearly stable.

**Illustrative steady states**

We present numerical results that illustrate the behavior of the APLG, including direct simulation of the particle model by the Gillespie algorithm[40], and time-stepping the (deterministic) partial differential equation (1) (see Methods). We consider fairly small domains so that the particle model simulations are tractable. [The total number of particles is approximately $(\ell_y/\ell_x)\phi(L/h)^2$, such that the simulations of Figs. 2 and 3 involve thousands of particles.]

**PS solutions.** The binodal construction (Fig. 1) demonstrates the existence of stationary solutions to (1) for large system size $L$. These consist of large $O(L)$ liquid and vapor regions, separated by interfaces of $O(1)$ width. Such PS states are familiar in NRCH, when the nonreciprocity parameter is not too large. In the current setting, the phase separation is caused by the particles' self-propulsion together with their excluded volume interactions: this is an example of MIPS[20,41–43], which has been previously characterized in the pure active case ($\phi_p = 0$) of this model[24].

Figure 2 shows the results of particle-based simulations of the APLG and corresponding numerical solutions of (1) in a domain of size $L = 2$. Phase separation occurs in both cases, starting from homogeneous initial states. We find that the dense phase is dominated by active particles, while the dilute phase is mostly passive. In fact, denser phases

always have lower concentrations of passive particles because of (4). As usual for MIPS, the magnetization $m$ is large in the interfacial regions but very small within the phases. Note that while the analytic binodal computation shows that stationary PS states exist, these numerical results also show that they are stable (for our choice of parameters).

**CP and traveling (T) solutions.** We now turn to dynamical steady states, which appear in the (blue) region D in Fig. 1b. We focus on two types: CP solutions retain an overall left-right symmetry with clusters of particles moving in both directions; T solutions break left-right symmetry, leading to a density profile that travels at a fixed speed. Specifically

$$\rho_\sigma(x, t) = \varrho_\sigma(x - ct/L), \qquad (6)$$

where $c$ is a constant, so the wave velocity is $c/L$; the reason for this $L$-dependence will be discussed below. (There is an analogy of CP and T solutions with standing waves and traveling waves, respectively, but note that both CP and T solutions are strongly anharmonic.)

An example CP state is shown in Fig. 3, which again compares direct simulation of the particle model with the numerical solution of (1), whose initial condition is a homogeneous state with a uniform random perturbation. This perturbation grows via an instability that involves CP sinusoidal waves with equal speeds and growth rates. After this transient growth, the resulting time-periodic state consists of two oppositely polarized active clusters that move in opposite directions. As they move, they accumulate passive particles in front of them via a "snowplow effect". The clusters collide, they pass through each other, and the cycle continues (see Supplementary Information Sec. F for additional discussion). In the example of Fig. 3, clusters appear to accelerate during collisions; a simpler example of this effect was observed in ref. 44.

Time-periodic states similar to CP have been observed in other active and nonreciprocal systems[11,45–47]: their presence here

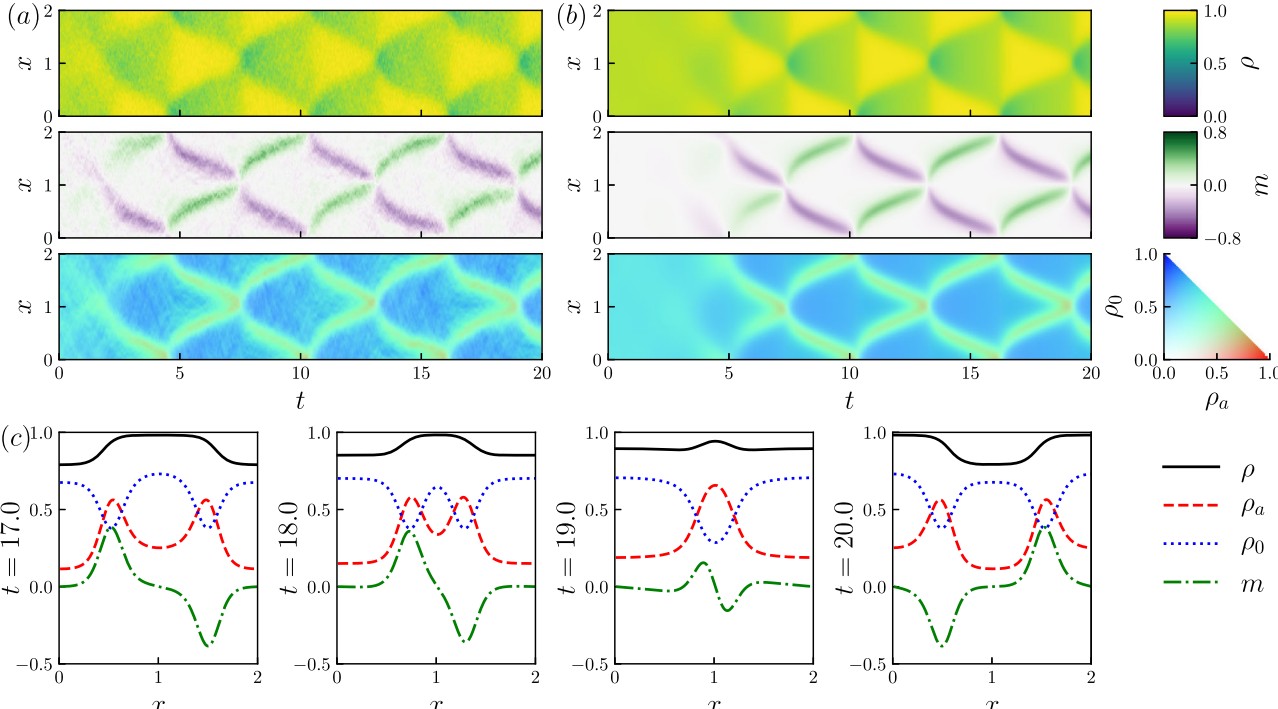

**Fig. 3 | Counter-propagating solution. a, b** Kymographs showing the spatio-temporal dynamics of a CP solution. **a** The $y$-averaged local density of a particle simulation (see Methods). **b** The solution to (1) whose initial condition is a homogenous state with a uniform random perturbation (see Supplementary Information Sec. E1, Eq. (S62)). **c** Density profiles before, during, and after the collision of two counter-propagating interfaces ($t = 17, 18, 19, 20$). Parameters: Pe = 20, $L = 2$, $\phi_a = 0.3$, $\phi_p = 0.6$, $\Delta x = 0.01$, $h = 0.01$.

emphasizes that they are generic. In particular, the APLG still supports the rich dynamical phenomenology of other active-passive mixtures, despite its idealized features. The connection between the APLG and other nonreciprocal systems is discussed further in Supplementary Information Sec. B. Despite the analogy with standing waves, we emphasize that these CP clusters experience complex (nonlinear) scattering processes when they meet (see, for example, Fig. S5).

An illustrative T state is shown in Fig. 4, obtained by time-stepping (1) for a larger system ($L = 25$). After an initial transient that resembles a CP solution, we find a solution of the form of (6). As in the CP case, this consists of a localized packet of active particles that pushes passive particles in front of it (see Supplementary Information Sec. F). We find numerically that such solutions can have a variety of shapes and non-trivial dependence on system size. To simplify this diverse behavior, we again turn to large systems ($L \gg 1$), which enables analytic progress, by analogy with PS states. Hence, we reveal new connections between dynamical patterns in this system and equilibrium phase coexistence.

### Traveling solutions in large systems

Recall that $L \gg 1$ corresponds to $\ell_x \gg \sqrt{D_T/D_R}$: this means that the system size is much larger than the intrinsic length scale associated with particle motion. Some active-matter and nonreciprocal models[9,48] support traveling PS states where a large system has macroscopic liquid and vapor domains separated by sharp interfaces, which move at constant velocity. However, this is not possible in the APLG. To see this, substitute (6) into (1) and write $z = x - ct/L$ to obtain

$$-(c/L)\varrho'_\sigma = \partial_z \left[ d_s(\varrho)\varrho'_\sigma + \varrho_\sigma \mathcal{D}(\varrho)\varrho' \right] - \text{Pe}\, \partial_z \left[ \varrho_\sigma s(\varrho)m + \sigma d_s(\varrho)\varrho_\sigma \right] - \sigma m,$$
(7)

where $\varrho_\sigma$ and $m = \varrho_+ - \varrho_-$ denote, respectively, the densities and magnetization in the traveling frame [recall (6)], and primes indicate derivatives with respect to $z$. Within the bulk of the phases, $\varrho' = 0$ so $m = 0$ in the bulk. Summing (7) over $\sigma$ to obtain the total density $\varrho$ and

integrating then yields $(c/L)\varrho + \varrho' - \text{Pe}(1 - \varrho)m = J$ where $J$ is an integration constant. Evaluating this expression inside the two phases, where $m = 0 = \varrho'$, one finds $\varrho = LJ/c$, so both phases would need to have equal densities, ruling out any traveling PS states (the special case $c = 0 = J$ recovers the PS state). This exact analysis illustrates the value of our exact hydrodynamic description: it is very difficult to extrapolate such results for large systems based on numerical simulations alone, especially because time-stepping Eq. (1) is expensive in large domains.

There are macroscopically smooth T solutions of (7) satisfying $\varrho'_\sigma = O(1/L)$ [with $c = O(1)$ and $m = O(1/L)$]. We do find such solutions numerically (see Methods). However, Fig. 4a includes an interfacial region where $\varrho$ varies quickly in space, hinting at the existence of solutions with traveling narrow interfaces [$\varrho' = O(1)$].

These T states with sharp interfaces do indeed exist. We find them systematically using the method of matched asymptotic expansions[49], with results shown in Fig. 5. This method consists of expanding solutions in an asymptotic series in the small parameter $1/L$: this is a controlled approximation scheme that provides accurate results for large systems. Specifically, we seek a solution to (7) with an interface at $z = 0$, and we split the domain of $z$ into two overlapping subdomains, an inner region around the wave front where $|z| = O(1)$ and an outer region far from the front ($|z| \gg 1$). Details of the calculation are given in Methods. In the inner region, the problem reduces, at leading order in $1/L$, to the same Eqs. (9), (10), (11) that govern PS solutions. That is, the leading order of sharp interfaces in T states coincide with liquid-vapor interfaces in PS, and connect points on the binodal by tie lines (recall Fig. 1). In the outer region, we can eliminate $m$, and the system reduces to two equations

$$-(c/L)\varrho'_a = \partial_z \left[ d_s(\varrho)\varrho'_a + \varrho_a \mathcal{D}(\varrho)\varrho' \right] + \frac{\text{Pe}^2}{2} \partial_z \left\{ \left[ \varrho_a s(\varrho) + d_s(\varrho) \right] \partial_z \left[ d_s(\varrho)\varrho_a \right] \right\}, \quad (8a)$$

$$-(c/L)\varrho'_0 = \partial_z \left[ d_s(\varrho)\varrho'_0 + \varrho_0 \mathcal{D}(\varrho)\varrho' \right] + \frac{\text{Pe}^2}{2} \partial_z \left\{ \varrho_0 s(\varrho) \partial_z \left[ d_s(\varrho)\varrho_a \right] \right\}, \quad (8b)$$

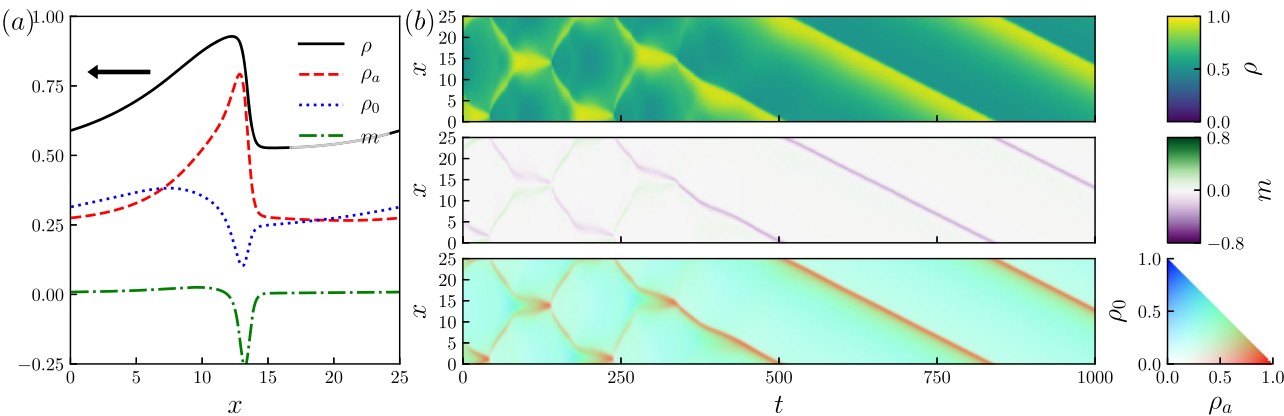

**Fig. 4 | Traveling solution.** Traveling profile solution to (1), whose initial condition is the homogeneous state perturbed by a combination of left and right traveling unstable sinusoidal modes and a uniform random perturbation (see Supplementary Information Sec. E1, Eqs. (S62, S63) for further details). **a** Density profile at $t = 500$. **b** Kymographs showing the spatiotemporal dynamics. Parameters: Pe = 7.5, $L = 25$, $\phi_a = 0.36$, $\phi_p = 0.3$, $\Delta x = 0.05$. Speed of propagation in (6) is $c = -1.8832$.

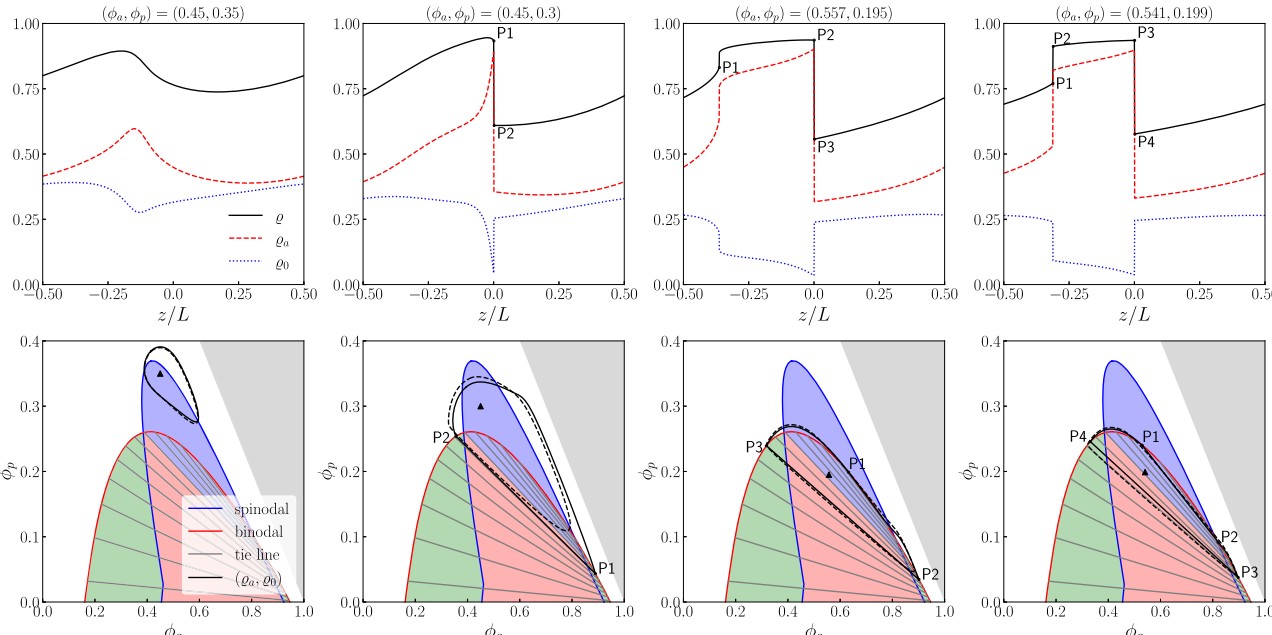

**Fig. 5 | Traveling solutions in the large system size limit.** T solutions of the finite-$L$ problem (7) and the large-$L$ problem (8) at Pe = 7.5. (Top) Solutions to (8) as a function of $z/L$. (Bottom) Solutions plotted parametrically on the phase diagram of Fig. 1b, with dashed lines for solutions of (7) and solid lines for solutions of (8). Volume fractions $(\phi_a, \phi_p)$ for each column are given in the top, also marked as black triangles in the bottom row. Other parameters: $N = 1024$ for numerical solution of (8) and $L = 25$, $N = 1600$ for (7). Wave speeds for (**a**–**d**) are $c = 1.95, 1.39, 0.41, 0.31$ (2 decimal places).

where $\varrho_a = \varrho_+ + \varrho_-$. Eq. (8) is to be solved with periodic boundary conditions at $z = \pm L/2$ and matching conditions to the binodal densities as $z \to 0^\pm$.

We numerically solve (8) for the densities $\varrho_a, \varrho_0$ and the speed $c$ and show the results for different combinations of $(\phi_a, \phi_p)$ in Fig. 5. The top row of Fig. 5 shows four solutions of (8), and the bottom row shows the same solutions overlaid on the phase diagram, together with the corresponding solutions to the original finite-$L$ problem (7). We see that the finite-$L$ solutions follow the tie lines, similar to the matched asymptotic solutions. Figure 6 shows how direct solutions of (7) approach the asymptotic solution as $L$ is increased, confirming that the matched asymptotic analysis provides accurate results. Further, the spinodal curves of the outer problem (8) agree with the spinodals of the full problem (see Supplementary Information Sec. D3).

It is in the outer region formulation (8) that the effective non-reciprocal interactions become most clearly apparent in the hydro-dynamic equation (1). In particular, we show in Supplementary Information Sec. B that (8) can be expressed as a nonlinear cross-diffusion system for the active and passive densities $\rho_a, \rho_0$ with non-reciprocal effective interactions between them.

We identify different types of T solutions by varying the volume fractions $\phi_p, \phi_a$. For pairs $(\phi_a, \phi_p)$ in the upper part of region D, we observe smooth (periodic) T solutions with no inner region, see Fig. 5a. Reducing $\phi_p$ in the phase diagram, we find solutions with a single interface (Fig. 5b). The inner region occurs between points P1 and P2 and lies along a tie line between two points on the binodal. Another interesting case is when the high-density part of the solution enters the binodal, leading to T solutions with two interfaces. Figure 5c shows this transition point: the first inner region is a single point (P1), tangent to

the binodal at its critical value, and the second interface occurs between P2 and P3. Slightly reducing $\phi_a$, we obtain solutions with two interfaces (Fig. 5d): these have two inner regions that both follow tie lines.

The array of solutions in Fig. 5 illustrates the rich phenomenology of the APLG in large domains. In particular, the appearance of narrow interfaces in T solutions is a surprising feature, since it connects these dynamically patterned states to the equilibrium-like constructions of the binodal and the associated interfacial profiles. While the analytical characterization of T solutions is not possible for CP solutions, numerical simulations show that CP solutions also feature narrow interfaces that follow the tie lines (see Supplementary Information, Fig. S5). The distinction between narrow interfaces and macroscopically smooth profiles would be very challenging to characterize by direct numerical solution of (1): the method of matched asymptotics makes this possible.

### Multistability

We have characterized T solutions, but this does not guarantee that the time-dependent system will converge to a T state, nor even that they are locally stable. There could also be other T solutions with different speeds $c$. To address this question, Fig. 7 shows the types of some long-time solutions obtained by numerically time-stepping (1).

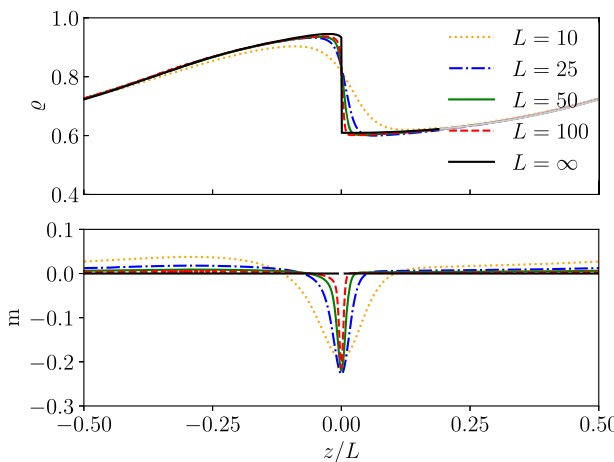

**Fig. 6 | Domain-size dependence of traveling solution.** T solutions of the finite-$L$ problem (7) for different values of $L$ and the large-$L$ problem (8) at Pe = 7.5. Total density $\varrho$ (top) and magnetization m (bottom). The magnetization for $L = \infty$ is m = 0 for $|z| > 0$. Parameters as in Fig. 5b.

The system is initialized with a zero-magnetization homogeneous state, perturbed by the most unstable eigenmode of the linear stability analysis. In the case of complex eigenvalues, solutions in Fig. 7a use a linear combination of left- and right-moving modes, while Fig. 7b uses only the left-moving mode. These systems converge to steady states, which we characterize as H, PS, T, or CP (see Methods for further details).

The results are consistent with Fig. 1 and demonstrate the existence and stability of T and CP states, for suitable parameters. Comparing Fig. 7a, b, the steady state also depends on the initial conditions: initializing with a left-moving mode favors T solutions while symmetric initialization favors CP. Figure 7b also shows the range of parameters over which we were able to find T solutions via matched asymptotics, showing that nonsymmetric initialization may not be sufficient to drive the system into T states, even if they exist. Together, these observations demonstrate multiple dynamical attractors, as may be expected for such complex PDE systems. Fig. S6 (in Supplementary Information) shows an explicit example where both T and PS solutions exist for the same parameters. Understanding the basins of attraction of different states in more detail is an important challenge for future work.

## Discussion

We introduced the APLG as a microscopic model of interacting particles and characterized its hydrodynamic behavior. In addition to PS states familiar from pure active systems, we find rich behavior, including that the spinodal curve can protrude through the binodal. This signals the existence of dynamical steady states, which we classify according to their symmetries. Some of these results are similar to previous work on the NRCH equation[9–11,13], but our hydrodynamic PDE includes the magnetization $m$ as a slow hydrodynamic variable, in addition to the two conserved densities $\rho_a$, $\rho_0$; it also has a distinct set of nonlinearities. Hence, our approach of deriving hydrodynamic equations exactly from a microscopic model complements the generic description of nonreciprocal systems via the NRCH equation. As already noted above, it is important that the APLG supports the complex behavior of other nonreciprocal models, despite the idealized modeling assumptions in its definition.

### APLG phenomenology

To tame the complexity of the APLG's behavior, we focused on large domains $L \gg 1$. This enables numerically exact computation of the binodal and spinodal curves and precise characterization of traveling solutions that involve sharp interfaces. Surprisingly, interfacial profiles in static and traveling states both obey (4), which also describes the tie lines in the phase diagram. If such connections are generic in

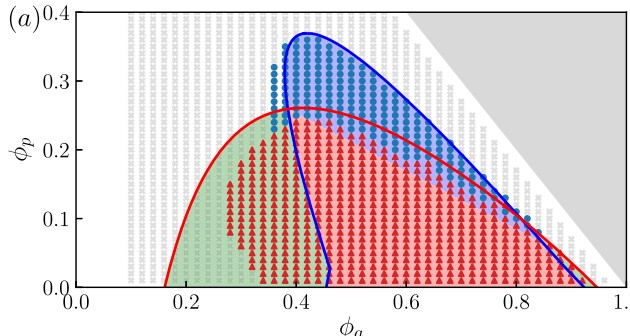

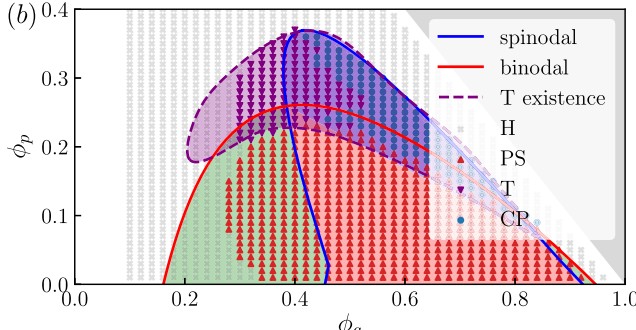

**Fig. 7 | Stability of T and CP solutions.** Phase diagrams from Fig. 1b, overlaid with the types of long-time solutions of (1): homogeneous (H), phase-separated (PS), traveling (T), and counter-propagating (CP), see Methods for the classification. The initial condition is the homogeneous state, perturbed by (**a**) left and right traveling unstable sinusoidal modes (**b**) left traveling unstable sinusoidal modes (see Supplementary Information Sec. E1 for details). Parameters: Pe = 7.5, $L = 25.0$, $\phi_a = 0.36$, $\phi_p = 0.3$, $\Delta x = 0.05$.

nonreciprocal systems, they would have broad consequences for understanding their phase diagrams, including possibilities for long-ranged order similar to equilibrium. We also demonstrated multi-stability: qualitatively different solutions can exist for the same parameters.

Our results also raise further questions for the APLG, including the existence of dynamical states in large systems with patterns on finite length scales, and associated questions of wavelength selection[11,13,50,51]. While this work analyzed deterministic hydrodynamic equations, the theory of fluctuating hydrodynamics can also be developed for such models[18,52,53] by retaining corrections to the limit $h \to 0$. This enables studies of metastability and the role of noise in determining a system's eventual steady state. There might also be interesting corrections to the large-$L$ behavior studied here. One may also expect new behavior on replacing the two-state active-particle orientation ($\sigma = \pm 1$) with a continuous degree of freedom: hydrodynamic limits can be derived in this case[19,25] but the resulting equations are challenging to analyze. Future work should investigate these issues.

### Implications for nonreciprocal systems

To provide a broader perspective, we note that the APLG model serves as an example of a general class of nonequilibrium mixtures. Its exact hydrodynamic limit relies on specific modeling assumptions, including a nontrivial $h$-dependence of the microscopic rates. However, the emergent large-scale picture is generic, including the appearance of complex stability eigenvalues, as observed in the NRCH and other nonreciprocal systems. As usual in mixtures of active and passive particles[9,15,22], this is driven by nonreciprocal effective interactions that appear in their hydrodynamic descriptions, notwithstanding that the microscopic interactions between particles are reciprocal.

For the APLG, we find that the protrusion of the spinodal through the binodal introduces a novel mechanism by which nonreciprocal behavior can arise, and it would be interesting to explore additional examples of this phenomenon. This could be achieved by considering hydrodynamic limits for other microscopic models, such as mixtures of different types of active particles[54]), or within the framework of field theories like the NRCH, independent of their underlying microscopic descriptions. For a recent example of an active system where the spinodal protrudes through the binodal, see refs. 55, 56.

An important takeaway from this work is the tractability of the large-system limit. Large systems are essential for understanding equilibrium phase behavior, and it is valuable to identify none-quilibrium situations where such systems can be analyzed precisely. In the case of the APLG model, this is facilitated by the presence of domain walls, which appear both in static phase separation and in dynamical steady states. The scale separation between interfacial width and system size allows this behavior to be tackled using the method of matched asymptotics, providing a new route for characterizing dynamical pattern-forming states, such as those observed numerically in refs. 15,22,54. We look forward to future work in this direction, which holds promising opportunities to bridge theories of pattern formation with those of equilibrium phase behavior (see also Supplementary Information Sec. B).

## Methods
### Coexisting phases

For large systems, $L \gg 1$, stationary PS states contain large domains of liquid and vapor phases. The phases' (total) volume fractions are denoted $\phi_l$, $\phi_v$, which we compute by generalizing the method of refs. 25,36,57. Specifically, setting $\partial_t \rho = 0 = \partial_t \rho_0$ in (1) and integrating yields

$$J = \partial_x \rho - \text{Pe}\,(1 - \rho)m, \tag{9}$$

$$J_0 = d_s(\rho)\partial_x \rho_0 + \rho_0 \mathcal{D}(\rho)\partial_x \rho - \text{Pe}\,\rho_0 s(\rho)m, \tag{10}$$

where $J$, $J_0$ are integration constants. Setting $\partial_t m = 0$ in the equation of motion for $m$ we obtain

$$2m = \partial_x \left\{ d_s(\rho)\partial_x m + m\mathcal{D}(\rho)\partial_x \rho - \text{Pe}\left[s(\rho)m^2 + d_s(\rho)\rho_a\right] \right\}. \tag{11}$$

In the bulk of either phase, we have $\partial_x \rho_\sigma = 0$, so from (11), $m = 0$ there. Then, evaluating Eqs. (9), (10) in the bulk shows that $J = 0 = J_0$. Using this fact and eliminating $m$ between Eqs. (9), (10)), one obtains using (2) that $\partial_x \log[\rho_0/(1 - \rho)] = 0$ so $\rho_0 = \nu(1 - \rho)$ for some constant $\nu$. Integrating this equation over the whole domain, one obtains the value $\nu$ in (5). Then, combining Eqs. (4), (9), (11), we obtain a condition on $\rho(x)$ alone:

$$\partial_x g(\rho, \partial_x \rho, \partial_x^2 \rho) = 0, \tag{12}$$

where

$$g(\rho, \partial_x \rho, \partial_x^2 \rho) = g_0(\rho) + \Lambda(\rho)(\partial_x \rho)^2 - \kappa(\rho)\partial_x^2 \rho, \tag{13}$$

with $g_0$, $\Lambda$, $\kappa$ given in Eqs. (S32)–(S33).

The method of refs. 25,36,57 can now be applied directly to (12). Within the bulk of the coexisting phases one has $\partial_x \rho = 0 = \partial_x^2 \rho$, showing that $g_0(\phi_l) = g_0(\phi_v)$. In addition, (12) can be used to construct an effective free energy $\Phi$, from which $\phi_l$, $\phi_v$ can be fully determined by a common tangent construction, see Supplementary Information Sec. C for details. The compositions of the phases are then given by (4). A numerical implementation of this procedure yields the binodal curves in Fig. 1.

### Linear stability of homogeneous solutions

To analyze the stability of homogeneous stationary solutions of (1), we introduce a perturbation to $\rho_\sigma$ constant of the form $\delta A^\sigma \exp(\lambda t + iqxt)$ for $\delta \ll 1$, and $\lambda$ can be obtained as the eigenvalue of a $3 \times 3$ matrix. If $\text{Re}(\lambda) > 0$, then the perturbation grows, signaling that the homogeneous state is unstable. The spinodal is the boundary between the regions of stable and unstable homogeneous solutions. Full details are given in Supplementary Information Sec. D.

### Traveling solutions via the method of matched asymptotics

To analyze T solutions in large domains, we define $\epsilon = 1/L$ and seek a systematic approximation of (7) as $\epsilon \ll 1$ via the method of matched asymptotic expansions. Recall that $z \in [-L/2, L/2]$ and that, without loss of generality, the interface is centered at $z = 0$ (see Fig. 6). We assume there is only one interface; the generalization to multiple interfaces is straightforward.

We define the outer region $|z| \gg 1$, where we set $z = \epsilon^{-1}\bar{z}$ and define $\varrho(z) = \bar{\varrho}(\bar{z})$ and $m(z) = \bar{m}(\bar{z})$ so (7) becomes

$$-\epsilon^2 c\bar{\varrho}'_\sigma = \epsilon^2 \partial_{\bar{z}}\left[d_s(\bar{\varrho})\bar{\varrho}'_\sigma + \bar{\varrho}_\sigma \mathcal{D}(\bar{\varrho})\bar{\varrho}'\right] - \epsilon\,\text{Pe}\,\partial_{\bar{z}}\left[\bar{\varrho}_\sigma s(\bar{\rho})\bar{m} + \sigma d_s(\bar{\varrho})\bar{\varrho}_\sigma\right] - \sigma\bar{m}, \tag{14}$$

with periodic boundary conditions at $\bar{z} = \pm 1/2$. Expanding $\bar{\varrho}_\sigma$ and $\bar{m}$ in powers of $\epsilon$, $\bar{\varrho}_\sigma \sim \bar{\varrho}_\sigma^{(0)}(\bar{z}) + \epsilon \bar{\varrho}_\sigma^{(1)}(\bar{z}) + \cdots$ and $\bar{m} \sim \bar{m}^{(0)} + \epsilon \bar{m}^{(1)} + \epsilon^2 \bar{m}^{(2)} + \cdots$, we find that the leading- and first-order of (14) lead to $\bar{m}^{(0)} = 0$ and $\bar{m}^{(1)} = -(\text{Pe}/2)\partial_{\bar{z}}[d_s(\bar{\varrho}^{(0)})\bar{\varrho}_a^{(0)}]$, respectively. The $O(\epsilon^2)$ of (14) is

$$-c\bar{\varrho}_\sigma^{(0)\prime} = \partial_{\bar{z}}\left[d_s(\bar{\varrho}^{(0)})\bar{\varrho}_\sigma^{(0)\prime} + \bar{\varrho}_\sigma^{(0)}\mathcal{D}(\bar{\varrho}^{(0)})\bar{\varrho}^{(0)\prime}\right] - \sigma\bar{m}^{(2)}$$
$$- \text{Pe}\,\partial_{\bar{z}}\left[\bar{\varrho}_\sigma^{(0)}s(\bar{\varrho}^{(0)})\bar{m}^{(1)} + \sigma d_s(\bar{\varrho}^{(0)})\bar{\varrho}_\sigma^{(1)} + \sigma d_s'(\bar{\varrho}^{(0)})\bar{\varrho}^{(1)}\bar{\varrho}_\sigma^{(0)}\right], \tag{15}$$

using that $d_s(\bar{\varrho})\bar{\varrho}_\sigma \sim d_s(\bar{\varrho}^{(0)})\bar{\varrho}_\sigma^{(0)} + \epsilon d_s(\bar{\varrho}^{(0)})\bar{\varrho}_\sigma^{(1)} + \epsilon d_s'(\bar{\varrho}^{(0)})\bar{\varrho}^{(1)}\bar{\varrho}_\sigma^{(0)}$ and $\bar{m}^{(0)} = 0$. Eliminating $\bar{m}^{(2)}$ from (15) by adding them for $\sigma = \pm 1$ leads to

$$-c\bar{\varrho}_a^{(0)\prime} = \partial_{\bar{z}}\left[d_s(\bar{\varrho}^{(0)})\bar{\varrho}_a^{(0)\prime} + \bar{\varrho}_a^{(0)}\mathcal{D}(\bar{\varrho}^{(0)})\bar{\varrho}^{(0)\prime}\right]$$
$$- \mathrm{Pe}\,\partial_{\bar{z}}\left[\left(\bar{\varrho}_a^{(0)}s(\bar{\varrho}^{(0)}) + d_s(\bar{\varrho}^{(0)})\right)\bar{m}^{(1)}\right], \tag{16}$$

using again that $\bar{m}^{(0)} = 0$. Inserting the expression for $\bar{m}^{(1)}$ into Eqs. (15), (16)) and switching back to $z$ leads to (8) at leading order in $\epsilon$.

To solve (8), it remains to determine the boundary conditions as we approach the interface, or $|\bar{z}| \to 0$. In the inner region $z = O(1)$, define densities $\hat{\varrho}_\sigma(z) = \varrho_\sigma(z)$. They solve (7), together with the matching condition to the outer region, $\lim_{z \to \pm\infty}\hat{\varrho}(z) = \lim_{\bar{z} \to 0^\pm}\bar{\varrho}(\bar{z})$; these limits exist under the assumptions of a localized interface. At leading order in $\epsilon$, the LHS of (7) vanishes: this ensures consistency with the argument above that traveling PS states do not exist, and justifies the scaling of the speed as $c/L$ in (7). Hence, the leading-order inner solution solves the same equations as the PS state [Eqs. (9), (10), (11) with $J = 0 = J_0$], and interfaces in T states connect points on the binodal curve.

## Numerical methods

- *Particle model*: The APLG is a continuous-time Markov chain on a finite state space. We simulate it exactly with the Gillespie algorithm[40], initially placing a $\sigma$-particle on a lattice site with probability $\phi_\sigma$, where $\phi_\pm = \phi_a/2$, $\phi_0 = \phi_p$. In Figs. 2 and 3, the simulated domain has $\ell_y = \ell_x/4$; we plot the $y$−averaged values of the mesoscopic densities (see Eq. (S2)) with radius $r = 0.1$.

- *Time-stepping of Eq.*(1): We use a first-order finite-volume scheme in space and forward Euler with adaptive time-stepping in time to obtain time-dependent numerical solutions $\rho_\sigma(x, t)$ to the hydrodynamic PDE (1), building on the numerical scheme of refs. 25,43 (see Supplementary Information Sec. E1).

- *Classification of the steady states*: We classify solutions of (1) into Homogeneous (H), PS, Traveling (T), or CP using the following two metrics: the approximate speed $\tilde{c}(t) := \|\partial_t\rho_\sigma\|_2/\|\partial_x\rho_\sigma\|_2$ and the distance from uniform

$$d_H(t) := \left(\sum_\sigma \|\rho_\sigma(\cdot, t) - \phi_\sigma\|_2\right)^{1/2}. \tag{17}$$

We solve (1) until a final time $t^* \geq 700$ when one of the following conditions is satisfied (where $\mathcal{T}^* = [t^* - 500, t^*]$):

(i) $\sup_{t\in\mathcal{T}^*} d_H(t) < 0.05 \to$ H solution.
(ii) $d_H(t^*) \geq 0.05$ and $\sup_{t\in\mathcal{T}^*}\tilde{c}(t) < 0.01 \to$ PS solution.
(iii) $d_H(t^*) \geq 0.05$, $\tilde{c}(t^*) \geq 0.01$ and $\sup_{t\in\mathcal{T}^*}|\tilde{c}'(t)| < 10^{-5} \to$ T solution.
(iv) $d_H(t^*) \geq 0.05$, $\tilde{c}(t^*) \geq 0.01$ and $\sup_{t\in\mathcal{T}^*}|\tilde{c}'(t)| \geq 10^{-5} \to$ CP solution.

- *Traveling profiles*: The profiles $\varrho_\sigma(z)$ and speed $c$ in Fig. 5 are obtained numerically by discretizing Eqs. (7), 8) with second-order centered differences and solving the resulting zeroth-finding problem subject to mass constraints $\int\varrho_0 dz = \phi_p$ and $\int\varrho_+ dz = \int\varrho_- dz = \phi_a/2$ in $[-L/2, L/2]$. The finite-$L$ system (7) is initialized with a preexisting T solution with similar parameters or a steady state of (1) and solved subject to periodic boundary conditions and $\varrho(0) = \phi$ without loss of generality (since the problem has translational symmetry). For $L \gg 1$, the numerical procedure to solve (8) is as follows. Given parameter values $\phi$ and $v$(5), we determine the inner solution via the abovementioned Coexisting Phases procedure. This results in liquid $(\varrho_\sigma)_l$ and vapor $(\varrho_\sigma)_v$ values, which become boundary values for the outer problem (8). [If $\phi_v(v) = \phi_l(v)$, it means that there is no inner region and we may proceed to solve (8) subject to periodic boundary conditions as in the finite $L$ case.] Then (8) is discretized using

second-order finite-differences and solved in $[0, L]$ subject to Dirichlet boundary conditions $\varrho_\sigma(0) = (\varrho_\sigma)_v$ and $\varrho_\sigma(L) = (\varrho_\sigma)_l$ and mass constraints as above. If no solution is found, it indicates there may be a second interface. We initialize with a previous single interface T solution with similar parameters. We then place a new interface at $z = \mathrm{argmax}_z|\partial_z\varrho|$. In both finite $L$ and $L \gg 1$ cases, we solve the resulting systems of equations using the `Nonli-nearSolve.jl` package in Julia (see Supplementary Information Sec. E2).

## Data availability

The simulation data generated in this study have been deposited in the Figshare database under accession code https://doi.org/10.25446/oxford.28881329[62].

## Code availability

The code used in this study has been deposited in the GitHub repository under accession code mbruna/Nonreciprocal_APLG[63].

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

## Acknowledgements
We thank Tal Agranov, Martin Burger, Mike Cates, Clement Erignoux, Sarah Loos, and Johannes Zimmer for helpful discussions. M. Bruna was supported by a Royal Society University Research Fellowship (grant no. URF/R1/180040). J. Mason was supported by the Royal Society Award (RGF/EA/181043) and the Cantab Capital Institute for the Mathematics of Information of the University of Cambridge. R.L. Jack was funded in part by EPSRC through grant EP/Z534766/1. For the numerical work, we used the Julia programming language[58] and the following packages and tools: DrWatson.jl[59], DifferentialEquations.jl[60], NonlinearSolve.jl[61]. For the purpose of open access, the authors have applied a CC BY public copyright licence to any accepted manuscript arising from this submission.

## Author contributions
J.M., R.L.J., and M.B. designed research, performed research, and wrote the paper.

## Competing interests
The authors declare no competing interests.
