## [Transparent Peer Review file · Nature Communications]

Dynamical patterns and nonreciprocal effective interactions in an active-passive mixture through exact hydrodynamic analysis

Corresponding Author: Professor Maria Bruna

Version 0:

Reviewer comments:

Reviewer #1

(Remarks to the Author)

The manuscript studies an active-passive model where the interactions are non-reciprocal. In contrast to previous studies of closely related systems, here the model is formalized in a specific scaling limit which allows for exact hydrodynamic equations to be derived. The most interesting finding is the phase diagram which displays a spinodal curve which crosses the binodal curve signaling an onset of pattern formation. The agreement between the analytical results and the numerics is convincing.

The paper, with good reason, prides itself of deriving exact results. Unfortunately many details remain rather vague. Even the exact scaling limit is unclear.

Detailed comments:

1. It will be useful to spell out what the authors mean by non-reciprocal interactions. The only real interactions are hard core.
2. In section I it is stated that the model is an extension of references [21-23]. However, the formulation there rescales the rates with system size while here the choice is a bit odd here and hidden with the scale h . This should be clarified. Namely, state explicitly the meaning of weak drive. What about D_R ? In Ref. 22 the flipping rate is scaled by the system size square, is it finite here?
3. Similarly, I would expect the $h \rightarrow 0$ limit to be related to a limit taken of the system size. This again has to be clarified. The question percolates everywhere in the paper, for example when the width of the domain walls is discussed or when the large L limit is taken.
4. Along the same lines, assuming I understand, it is stated that the model is non-gradient and therefore mean-field does not hold. However, intuitively is the diffusion is the dominating behavior and therefore I would expect the leading order measure to be a product measure. The discussion has to be expanded to make the paper clear.
5. Similarly, there is no real derivation of the hydrodynamic equations in the supplementary material. It just refers the reader to a paper which is not easy to parse. It will be useful to give the gist of the method used. Is the dimension 2 of any importance, or will the same results hold in a 1d version of the model?
6. It is not clear in equation 3 what type of approximation is made.
7. It will be nice, if possible, to have a simple intuitive picture of the CP phase with the specific interactions used in the model. At the moment it is spelled out as a results. The same comment holds for the T phase.
8. In the section discussing multi-stability, would it not be needed first to consider the (weak) noise terms to conclude which state is more stable?

In summary, while the phase diagram derived in the paper is very interesting from a statistical physics point of view the methodology and details needed to understand the paper fully are missing. Moreover, the authors do not make a clear case of what phenomenology is new when compared to previous works such as the non-reciprocal Chan-Hilliard equation. This also makes it unclear to me whether the place of the paper in Nature Communications which is aimed at a broad audience. However, I leave the authors to option to convince me otherwise.

(Remarks on code availability)

Reviewer #2

(Remarks to the Author)

Dear editor,

In the submitted manuscript the authors study an active-passive lattice gas model. As in several recent works in active matter, the rates are scaled in a way that there is a limit in which it becomes equivalent to a continuum equation derived by the authors. They study the different phases of this system: the phase-separated state (as seen in the fully active case) and patterns seen only in the mixture (traveling state and counter-propagating patterns). They report on the interesting possibility that the spinodal protrudes out of the binodals due to the non-reciprocal couplings at hydrodynamic level.

The method used proved itself, especially in active matter with Ref.[22,23,40] (the preprint Martin et al, 2307.08251 should probably be cited too). I found the paper well written and the results seem solid, with the small reservation explained below. I thus think this manuscript deserves publication in Nat Comm after the comments below are answered.

- The scaling used allows to derive exact continuum equations but at the price of having unphysical rates in the model. As I see it, it is akin to choosing the rates so that the mean-field equations become exact. I am not suggesting that authors do a full study of fluctuations but the fact that the results are in essence mean-field should be acknowledged in some way.

- I find section IV to be rather weaker than the rest of the paper. For someone not familiar with the "multi-scale matched asymptotic method", it feels like a bit of "cooking" to extract a solution with uncontrolled approximations. I am ready to be convinced that the extracted solution is close to the real one but this is not shown in the paper. I believe the authors should at least compare the solutions obtained with this method and the direct integration, for example with the traveling states seen in Fig.6b.

One issue that I have is that in similar situations when one is looking for non-linear solutions (e.g. FKPP fronts or, perhaps closer to the present topic, the patterns seen in flocking models as in Solon et al, Phys. Rev. E 92, 062111 (2015)), there exists, for the same parameters, a family of patterns with different velocities. Here the authors suggest that the velocity is selected. Can they comment on why and how?

- I tried to figure out how many particles there are in the simulations of Fig.2 but could not based on available parameters (one cannot infer l_x only from Pe). Please indicate all the parameters.

- p6: what are "natural" solutions of (7) ?

- p6:

>> some active-matter and non-reciprocal
>> models [9, 40] support traveling phase-separated states
>> ("traveling bands")

Citing Ref.40 here is rather strange as it comes after many papers on traveling bands in flocking models: the first bands in the Vicsek model are seen in Chaté et al, Phys. Rev. E 77, 046113 (2008), the first traveling phase separated state in the Active Ising model in Solon and Tailleur, PRL 111, 078101 (2013).

The formulation "traveling phase-separated states ("traveling bands")" is also unfortunate because "bands" (an extensive number with a finite

size, as in the Vicsek model) are usually distinguished from phase-separated states (of size proportional to system size, as in the active Ising model).

(Remarks on code availability)

Reviewer #3

(Remarks to the Author)

The present work considers a mixture of passive and active particles in 2 spatial dimensions (2D).

The authors first provide a microscopic agent based model (Active Passive Lattice Gas Model), where the passive particles diffuse while the active particles self-propel (in the x-direction only to allow for a later thorough analytic treatment) according to a discrete Poisson variable (left/right). Passive (resp. active) particles remain passive (active) during all the simulation.

The authors then build on previous results in the literature to derive an hydrodynamic continuous model. The results of the simulations for the microscopic and hydrodynamic models are compared: they are qualitatively and quantitatively similar. The authors investigate the phase diagram in terms of the active/passive densities, showing that the binodal and the spinodal intersect. This opens the door to interesting dynamical phenomenology, here propagating waves, a ubiquitous feature in non-reciprocal active matter. In particular, they describe the dynamics of two similar phenomena: Counter-propagating (CP) and Traveling (T). T waves are isolated density fronts that propagate unidirectionally, without perturbation. CP waves, if we got it correctly, result from the interaction/collision of 2 T waves travelling in opposite directions.

The authors used a multi-scale method of matched asymptotic expansions to obtain the profiles (in space and time) of travelling waves.

The manuscript content is dense, complete and technical but remains understandable. Its main scope is to establish rigorous (generic) results based on a simple model that could eventually serve as a reference to understand the dynamical pattern formation behaviour of more complex active systems, where an analytical treatment is out of reach. I believe this objective is accomplished, although I wonder if Nature Communications is the right place for such fundamental, formal study, on a specific field within active matter physics. As one finds in the manuscript their "approach [...] complements the generic description of non-reciprocal systems via the NRCH equation". Besides that, the manuscript should be improved before publication, to clarify the points we list below.

Major comment on the non-reciprocity

In addition to the title, the authors refer to their model system as being 'non-reciprocal' and the relation with non-reciprocity is mentioned several times throughout the main text :

« Our exact hydrodynamic equation differs from NRCH, but the resulting phenomenology is similar and consistent with generic principles of self-organization via non-reciprocity. » (Mason et al., 2024, p. 1)

« This work addresses these challenges by analyzing a specific non-reciprocal system for which exact mathematical results can be derived » (Mason et al., 2024, p. 1)

« Our microscopic model clearly exhibits the principles of non-reciprocal self-organization. » (Mason et al., 2024, p. 2)

« In particular, (1) includes non-reciprocal couplings between active and passive densities. » (Mason et al., 2024, p. 3)

However, we don't understand what the authors mean by 'non-reciprocal' in their system. We don't see any source of non-reciprocity in the features of the dynamics written under Section I ACTIVE-PASSIVE LATTICE GAS (APLG) MODEL : (i) is pure diffusion, (ii) is self-propulsion, (iii) is excluded volume and (iv) is a Poisson process for the orientation of the active particles.

The authors suggest (quite vaguely) that non-reciprocity arises from the bidispersity, mixing passive and active particles. We don't understand why.

On the other hand, neither we clearly identify the source of non-reciprocity in Eq. (1).

Could the authors please detail this important point ? maybe providing their definition of non-reciprocity and stating explicitly where does it appear in their micro and continuum model ?

Major comment on the Counter Propagating waves

FIGURE 3:

This figure is important to understand the phenomenon of interest: the 2 counter-propagating waves.

However, it is not clear to us what happens precisely during the collision for the two clusters to leave with a higher speed/acceleration, before coming back to a constant speed (straight lines in the kymograph). Moreover, the authors write « The clusters slow down during collisions, but they eventually pass through each other » It looks like the opposite to us, they accelerate; higher slopes in the t-x plane means higher velocity. Or did we misunderstand something? Anyway, this point needs further clarification.

Also, as a side note on the choice of the colours of the first row: as the yellow usually is a background colour, we first thought that the cluster was in red. It is just a minor comment as the conveyed message remains clear, thanks to the other 2 rows and since the patterns of the high/low density regions are eventually the same due to the periodic boundary conditions.

Also, it is not clear to us why the CP propagating waves are « sinusoidal » (p. 5, left column). The profiles (Fig4c) seem more gaussian or lorentzian.

More generally :

The manuscript refers to the CP solutions and the T solutions as two distinct phenomenologies. How wrong is it to understand CP waves as a specific case of 2 single T waves ? Does the two waves constituting the CP solution have any additional properties compared to a single T wave ?

Also, even though it seems complex, do the authors have an intuitive understanding of what happens for $t < 250$ in Fig.4(b) ? How does it compare to the CP solution ?

Why does it collapse to a single T solution at one point, changing the symmetry of the pattern? Could this left/right spontaneous symmetry breaking also happen after a while in the true CP solution?

Comparing Figure 3 and Figure S3.

Figure 3 of the main text describes the CP solutions, composed of two waves. In this particular realisation, both waves are soft (smooth bumps in the density field). Figure S3 also describes a CP solution, the difference is that it seems to be a lot sharper, cf panel (a). An important scattering seems to occur in the kymographs of panel (b), could you explain why you don't observe this in Figures 3 and 4 of the main text?

Minor comments

1. The active particle density ρ_a defined below eq. 1 is not used.
2. The introduction of eq. 2 is grounded on the SEP: could the authors say a bit more on the origin of eq 2 in order to make the text somehow more self-contained?
3. The black line in Fig 1 b is not mentioned in the caption neither explained. The grey region in this panel is also quite obscure.
4. We wonder what would change if instead of having a passive-active mixture, one mixes particles with two different activity levels? Is there a qualitative difference? Is it the activity contrast the relevant parameter here, the density ratio, both? What is the role played by each one of these parameters? A discussion along these lines would help putting these results in the right context.
5. It is mentioned that MIPS "is also a familiar phenomenon in non-reciprocal matter". Here again a discussion is missing on what do the authors have in mind with non-reciprocity, as MIPS is typically referred to a phase transition emerging from the competition between self-propulsion and excluded volume, with no non-reciprocity (or at least not discussed with this jargon).
6. PRECISION :
« the mean-field approximation $\langle \eta \sigma(x, t) \eta \sigma(\hat{x}, t) \rangle \approx \rho \sigma(x, t) \rho \sigma(\hat{x}, t)$ does not hold and no explicit formula for $ds(\rho)$ exists. » (Mason et al., 2024, p. 2) This technical point is interesting. Does it mean that there are very strong spatial correlations in the system or do I get it wrong? Is it easy to provide an intuitive explanation?
7. « [Inside the spinodal, the homogeneous state is linearly unstable, and the phase-separated (PS) state is stable; between the spinodal and binodal, the PS state is globally stable while the homogeneous state is metastable]. » (Mason et al., 2024, p. 3) You can remove the brackets, this technical comment is useful to non-expert readers.
8. FIGURE 4: It would help the reader to add an arrow indicating the direction of propagation of the T propagation. I understand why there is a peak in the passive density in front of the profile (the passive particles are effectively « pushed » to the left by the active ones: the active ones want to go left so they don't move until the passive one diffuses to the left, and the passive particles can only diffuse to the left due to the combined effects of active particles being to their right + excluded volume. However, I don't think I understand why the active density exhibits such a sharp profile, which explains the majority of the total peak in density of those T waves. Could the authors please explain this point?
9. FIGURE 5: This good plot contains a lot of details. However, you might consider
 - increasing the size of the points P1 P2 P3 P4, we don't see them clearly even after a zoom, especially in the bottom row.
 - indicating the densities (ϕ_a , ϕ_p) used for each column on top of each top row panel. You have the space and it would help the reader to know straight away what changes from one column to the other.
 - changing the black cross symbol to another one, the black cross has been already used on the phase diagram (Fig. 1) to specify the bifurcation of co-dimension 2.
10. « Reducing ϕ_p » (Mason et al., 2024, p. 7) : you indeed reduce ϕ_p but you mostly reduce ϕ_a , to exit the Dynamical region...
11. SM Figure 3 :
« Density profile at $t = 500$ » (Mason et al., 2024, p. 19). In the title of panel (a) it's written $t = 4000$.
12. Typos « Parameters », below Eq (S41) and in the last paragraph: with parameters $reltol=1e-8$, $abstol=1e-8$

(Remarks on code availability)

Reviewer #4

(Remarks to the Author)

(Remarks on code availability)

Version 1:

Reviewer comments:

Reviewer #1

(Remarks to the Author)

I have read the new manuscript and find it to be much improved. Specifically, the protrusion of the spinodal through the binodal is emphasized - a result that is likely to be found to be important in other systems. Other comments have also been clarified. I therefore recommend publication.

(Remarks on code availability)

Reviewer #2

(Remarks to the Author)

I am satisfied by the reply of the authors and thus recommend publication.

(Remarks on code availability)

Reviewer #3

(Remarks to the Author)

The authors have addressed the points raised in our previous report in a satisfactory way. I now understand better the scope of the work and its fit in Nature Communications. Regarding the 'non-reciprocity' arising in the model, I still have a few doubts that I believe should be clarified before publication. As the work is motivated from the viewpoint of non-reciprocal interactions, this concept has to be as clear and rigorous as possible.

In the discussion SM sec B 1, one needs to assume that the friction matrix is independent of the system's micro-state. But then it is said that $\Gamma_0 = \Gamma(0)$. It is confusing, because otherwise one would get also gradients of Γ in eq. S6. As mentioned after S9, the friction matrix generically depends on the configuration of the system/field. Regarding the non-reciprocal case discussion, for which the linearised equations or the matrix, is no longer symmetric, is also somehow unclear. Newton's third law establishes the equivalence between forces between bodies, not their gradients, as concluded by S8. $A_{\{NR\}}$ would be the gradient of the friction and force, and should also be written explicitly. Might be also useful to provide a simple example as the predator-prey problem mentioned, or a mixture of active-passive particles. Do we need to go to a coarse-grained description to see non-reciprocity in active particle systems?

The discussion about the PDE system is very useful indeed. But the 'mechanistic' particle-based one, is not.

From this discussion I conclude that non-reciprocity arises from having a mixture of active and passive particles, at the continuum level. I guess this is more general, and any mixture of active particles with different activity would also be non-reciprocal in this sense (we don't need passive particles in the model, but just more than one species).

(Remarks on code availability)

Reviewer #4

(Remarks to the Author)

(Remarks on code availability)

Version 2:

Reviewer comments:

Reviewer #3

(Remarks to the Author)

The authors have clarified all the points I raised in my previous report. The article has improved, and in my opinion it is now ready for publication.

(Remarks on code availability)

Reviewer #4

(Remarks to the Author)

I co-reviewed this manuscript with one of the reviewers who provided the listed reports. This is part of the Nature

Communications initiative to facilitate training in peer review and to provide appropriate recognition for Early Career Researchers who co-review manuscripts.

(Remarks on code availability)

NCOMMS-24-47104

Title: Dynamical patterns and nonreciprocal effective interactions in an active-passive mixture through exact hydrodynamic analysis

Authors: J. Mason, R. L. Jack and M. Bruna,

We thank the referees for their careful reading of our manuscript and their constructive comments.

Below, we provide a summary of the main changes and a point-by-point response to the specific comments raised by each referee. The number references (to bibliography, equations, or figures) in the referee reports and our responses correspond to those in the revised manuscript. For convenience, the revisions to the manuscript are highlighted in red.

Summary of main changes

- We changed the title of the paper to be consistent with Nat Comms guidelines on punctuation/formatting.
- Three referees asked about the nonreciprocal interactions in our system. We clarify here that we refer to *effective interactions* that appear in the hydrodynamic equations. We have changed “interactions” to “effective interactions” in the title; we added a new Section B in the Supplementary Information to further explain our use of the “nonreciprocal” terminology; and we have highlighted the role of such effective interactions in the main text.
- We added a new Figure 6 to illustrate that the matched asymptotic analysis provides accurate results in large systems. We also adjusted some parameters in Figure 5 to better illustrate the convergence of the full solution to the matched asymptotic ones.
- We added a new Sec. VI.b to explain how the results for the (idealized) APLG model can inform future work on nonreciprocal systems in general. We also comment on this in the revised Introduction.
- We revised and extended Sec. A of the Supplementary Information to clarify how we take the hydrodynamic limit.
- We added Sec. F of the Supplementary Information to clarify how we expect counterpropagating CP solutions to behave in large systems and their relation to traveling solutions.

Detailed response

Referee #1

The manuscript studies an active-passive model where the interactions are nonreciprocal. In contrast to previous studies of closely related systems, here the model is formalized in a specific scaling limit which allows for exact hydrodynamic equations to be derived. The most interesting finding is the phase diagram which displays a spinodal curve which crosses the binodal curve signaling an onset of pattern formation. The agreement between the analytical results and the numerics is convincing.

The paper, with good reason, prides itself of deriving exact results. Unfortunately many details remain rather vague. Even the exact scaling limit is unclear.

We thank the referee for these constructive comments. Regarding the clarity of the scaling limit, we have edited Section I.a of the main text to clarify what we do. See also the detailed responses below. To briefly summarise: The dynamical rates of the particle model are described in the first paragraph of Sec I. We clarified that the relevant limit takes the lattice spacing $h \rightarrow 0$ at fixed $\ell_x, \ell_y, \phi_a, \phi_p$. This means that the number of lattice sites and the number of active/passive particles diverge in the limit. The relationship to the thermodynamic limit – which is relevant for finite-size scaling in physics – is discussed in a new SI Sec A.1. This new Section also explains why the dynamical rates of the particle model have nontrivial h -dependence.

1. *It will be useful to spell out what the authors mean by nonreciprocal interactions. The only real interactions are hard core.*

We thank referees #1, 3, 4 for making this point. A crucial point of our reference to non-reciprocity, which we should have emphasized more strongly, is the *effective* aspect of the nonreciprocal interactions. In particular, nonreciprocity emerges at the macroscopic level from simple exclusion interactions combined with self-propulsion at the microscopic level. We have clarified this point in the main text and added a new section in the SI (Sec. B) explaining what we mean by nonreciprocity and how our model fits within this framework. We have also amended the manuscript title to include the term “effective.”.

The nonreciprocal effective interactions can be derived exactly in particular regimes (see again Sec. B of SI). However, the behavior we observe – including complex stability eigenvalues and traveling steady states with one species “chasing the other” – are fingerprints of nonreciprocal self-organization and provide strong evidence that nonreciprocal effective interactions are present.

2. *In section I it is stated that the model is an extension of references [19,23,24]. However, the formulation there rescales the rates with system size while here the choice is a bit odd here and hidden with the scale h . This should be clarified. Namely, state explicitly the meaning of weak drive. What about D_R ? In Ref. 23 the system size square scales the flipping rate, is it finite here?*

We agree that this is an important point (also raised by referee 2). The model is defined exactly as in Section I of the main text, so the particle hop rate is proportional to h^{-2} and the orientational flip rate is independent of h . The drive is weak because the hop rate is

$$D_T/h^2 + \frac{v_0}{2h^2}(\mathbf{u} \cdot \mathbf{e}_\sigma)$$

with $|\mathbf{u}| = h$. So the driving term (proportional to v_0) is $O(h^{-1})$ which is much smaller in magnitude than the diffusive term (D_T/h^2).

We have revised both Sec. I.a and Sec. A of the SI to clarify that the hydrodynamic limit is $h \rightarrow 0$ at fixed domain lengths ℓ_x, ℓ_y , so the number of lattice sites diverges in this limit, as do the numbers of active/passive particles. As noted above, particle hop rates also depend nontrivially on h . These scalings are equivalent to those in [19,23,24]. (Making the connection to Ref. [23] requires some care: one should identify $h = 1/L$, the microscopic time unit in that work also differs by a factor of h^2 from ours, and their model also differs in that adjacent particles with different orientations can swap positions, whereas the APLG only allows particle hopping into vacant sites.)

For the presentation in this work, we did not follow exactly Ref. [23] because it is important for us that the inverse lattice spacing $1/h$ has a different physical meaning from the system size,

which we identify with ℓ_x (or $\ell_x \times \ell_y$). The number of lattice sites is $\ell_x \ell_y / h^2$, so it is true that the “hydrodynamic limit” $h \rightarrow 0$ corresponds to a diverging number of lattice sites, but this is not a classical large-system limit because the particle hopping rates depend on h . The physical large-system limit would be $\ell_x, \ell_y \rightarrow \infty$, which is similar to the “thermodynamic limit” in equilibrium systems. This “thermodynamic” limit is taken in Sec IV. The relationship between hydrodynamic and thermodynamic limits is discussed in Sec A.1 of SI.

3. *Similarly, I would expect the $h \rightarrow 0$ limit to be related to a limit taken of the system size. This again has to be clarified. The question percolates everywhere in the paper, for example, when the width of the domain walls is discussed or when the large L limit is taken.*

See previous comment. We have added a paragraph in Sec. I.a of the main text to clarify that the limit $h \rightarrow 0$ has some differences from a standard limit of large system size. This is related to the nontrivial h -dependence of the particles’ dynamics, which enables the exact mathematical analysis of this work. We have also added comments in Sec. A.1 of SI, explaining how this idealisation affects the results, and discussing how our results provide useful generic insights despite the idealised model.

For the discussion of domain walls, we have added an additional comment that the limit $L \gg 1$ in Sec. IV corresponds to $\ell_x \gg \sqrt{D_T/D_R}$, corresponding to a system that is much larger than the interfacial width. This is a separate limit from the hydrodynamic limit, which is $h \rightarrow 0$ with fixed D_T, D_R . So the discussion of Sec. IV is relevant for $h \ll \sqrt{D_T/D_R} \ll \ell_x$.

4. *Along the same lines, assuming I understand, it is stated that the model is non-gradient and therefore mean-field does not hold. However, intuitively is the diffusion is the dominating behavior and therefore I would expect the leading order measure to be a product measure. The discussion has to be expanded to make the paper clear.*

The referee’s intuition here is mostly correct, but additional subtleties exist for non-gradient models such as this. The “leading order” measure is indeed the standard product measure predicted by mean-field. However, sub-leading corrections to this measure are still large enough to affect the hydrodynamic limit equation.

In the rigorous theory of hydrodynamic limits, these corrections appear when one approximates a discrete flux of particles (on the lattice) by a function of the local density gradients. This happens by means of a replacement lemma, which applies in the limit $h \rightarrow 0$. Proving such replacement lemmas is technically challenging in non-gradient systems because corrections to the local product measure are hard to characterize in general. In our case, we are able to exploit a general replacement lemma for active systems, as proved by Erignoux [Eri21], see also [MEJB23].

As a simple example where mean-field does not hold, one may consider a two-species SEP as considered by Quastel [Qua92], in which relaxation of composition fluctuations are intrinsically linked to their self-diffusion constant. It can be proved (or checked numerically) that mean-field approximation gives the wrong result for the self-diffusion constant.

To understand this failure of mean-field, one may consider the problem satisfied by the two-particle probability density function $P_{\sigma, \sigma'}(x, y, t)$ of two particles of different types σ, σ' undergoing SEP. In [MJB23], we studied this problem to derive the macroscopic PDE of a two-species SEP formally but systematically in the limit of low volume fraction, $\phi \ll 1$ but finite. The mean-field approximation is $P_{\sigma, \sigma'}(x, y, t) = p_\sigma(x, t)p_{\sigma'}(y, t)$ but the systematic

expansion unveils that $P_{\sigma,\sigma'}(x,y,t) = p_{\sigma}(x,t)p_{\sigma'}(y,t) + h\psi_{\sigma,\sigma'}((y-x)/h) + O(h^2, \phi)$ when $|y-x| \sim h$, where ψ is a nontrivial function which only vanishes when $\sigma \equiv \sigma'$ (see Sec. 3.3 and Eq. (49) of [MJB23]). This nontrivial term, omitted in the mean-field approximation, is what gives rise to the α terms in our polynomial approximation of $d_s(\rho)$ in Eq. (3) of the main text (mean field would predict $\alpha \equiv 0$).

We have expanded the explanation in Sec. A of the SI to include these points.

5. *Similarly, there is no real derivation of the hydrodynamic equations in the supplementary material. It just refers the reader to a paper which is not easy to parse. It will be useful to give the gist of the method used. Is the dimension 2 of any importance, or will the same results hold in a 1d version of the model?*

The mathematical proof of the hydrodynamic limit is extremely technical and closely follows the derivation of [Eri21] (over 100 pages). In [MEJB23], we considered an active exclusion process identical to the one in the present paper and included an abridged version of the proof in Sec. 5 of [MEJB23]. We have added some further key points of the derivation as well as the reference to the abridged derivation in Sec. A of the SI.

The method works for any dimension equal to or bigger than two. The situation is different in one dimension because single-file diffusion is not sufficient to mix the system. In one dimension, one may prove the hydrodynamic limit of a simplified process where swaps in symmetric jumps are allowed (leading to linear diffusion terms in the hydrodynamic PDE, see [KEBT18, Eqs. (3,4)]. We have added a sentence in Sec. I to clarify this.

6. *It is not clear in equation (3) what type of approximation is made.*

The key point here is that we are approximating a real-valued C^∞ function of one variable, this is a simple task in practice and many methods are available. See Fig. R1 below where numerical data (points) is compared with our approximation (green line). These data are reproduced from [MJB23].

We have added a sentence below Eq. (3) to explain that our approximation matches exactly the values of $d_s(\rho)$ and $d'_s(\rho)$ at both ends $\rho = 0, 1$. We have also added a paragraph at the end of Sec. A in the SI to further detail the approximation method and how this approximation compares with the mean field result.

Figure R1 shows our approximation of $d_s(\rho)$ in green, compared with the mean-field approximation (red dash-dot line) and particle simulations (circles). It also shows the exact expansions of $d_s(\rho)$ around $\rho = 0, 1$.

7. *It will be nice, if possible, to have a simple intuitive picture of the CP phase with the specific interactions used in the model. At the moment it is spelled out as a results. The same comment holds for the T phase.*

We have added Sec. F in the Supplementary Information explaining this (including a sketch in Fig. S7 as suggested by the referee), and given some extra discussion of the CP state.

8. *In the section discussing multi-stability, would it not be needed first to consider the (weak) noise terms to conclude which state is more stable?*

In the multistability discussion of Sec. V, we meant local stability. We have corrected the text accordingly. From the hydrodynamic model, we cannot say anything about global stability;

Figure R1: Self-diffusion coefficient $d_s(\rho)$ in a Simple Size Exclusion Process (SSEP) in two dimensions reproduced from [MJB23]. Exact expansions $d_{s,0}(\rho)$ and $d_{s,1}(\rho)$ around $\rho = 0, 1$, respectively, and composite approximation $d_{s,c}(\rho)$ shown in Eq. (3) in the main text. For comparison, we also show the mean-field approximation d_s^{MF} and the results from particle simulations.

as pointed out by the referee, this would require adding noise.

In summary, while the phase diagram derived in the paper is very interesting from a statistical physics point of view the methodology and details needed to understand the paper fully are missing. Moreover, the authors do not make a clear case of what phenomenology is new when compared to previous works such as the nonreciprocal Cahn-Hilliard equation. This also makes it unclear to me whether the place of the paper in Nature Communications which is aimed at a broad audience. However, I leave the authors to option to convince me otherwise.

In response to this point, we have expanded the discussion of the manuscript to emphasize that our results have general importance, beyond the specific APLG model considered here. Specifically, the new phase diagram that we find in Fig. 1(b) illustrates a generic class of behaviour in nonequilibrium mixtures, that we expect to occur in a range of models. The fact that domain-wall solutions from static phase separation also appear in dynamical steady states is also a surprising new result that may occur more generally, offering an opportunity to rationalize the behaviour of many different nonequilibrium systems. These – and other – perspectives have been discussed in Secs. I, II, VI of the main text. We hope that these are sufficient to convince the referee of the broad potential audience for these results.

Referee #2

In the submitted manuscript the authors study an active-passive lattice gas model. As in several recent works in active matter, the rates are scaled in a way that there is a limit in which it becomes equivalent to a continuum equation derived by the authors. They study the different phases of this system: the phase-separated state (as seen in the fully active case) and patterns seen only in the mixture (traveling state and counter-propagating patterns). They report on the interesting possibility that the spinodal protrudes out of the binodals due to the nonreciprocal couplings at hydrodynamic level.

The method used proved itself, especially in active matter with Ref. [23,24,44] (the preprint Martin

et al, 2307.08251 should probably be cited too). I found the paper well written and the results seem solid, with the small reservation explained below. I thus think this manuscript deserves publication in Nat Comm after the comments below are answered.

We thank the referee for bringing to our attention the preprint by Martin et al., which considers an active spin model for two species with nonreciprocal alignment interactions and derives its fluctuating hydrodynamic limit. We have included it in our revised version (Sec. VI.a and Sec. A.1 in the SI).

1. *The scaling used allows to derive exact continuum equations but at the price of having unphysical rates in the model. As I see it, it is akin to choosing the rates so that the mean-field equations become exact. I am not suggesting that authors do a full study of fluctuations but the fact that the results are in essence mean-field should be acknowledged in some way.*

Re: “unphysical rates” (this point is related to comment 3 of referee 1). The referee’s comment applies to the h -dependent rates that appear in the definition of the model. This aspect is discussed in the revised sections I.a of the main text and A.1 of the SI. This h -dependence indeed enables our exact analysis. However, we do not agree that this h -dependence is unphysical, and we also note that our hydrodynamic equations are not those predicted by a (standard) mean-field analysis. Despite this, we do understand the referee’s point that the model is an idealized one. (One might make this comment based on the h -dependent rates, or because of the use of an underlying lattice, etc.)

In reply, we note that idealized models are common in mathematics and theoretical physics: the question is whether the modeling assumptions are justified and whether valuable conclusions can be drawn. In the current situation, we observe the following: (i) The phenomenology of the APLG captures many features of other nonreciprocal models, including the NRCH equation, as well as off-lattice particle models with dynamical steady states [WWG16, SWMC15]. Hence it does not seem that the modeling assumptions are too restrictive. (ii) The modeling assumptions enable new exact results for the phase diagram and the structure of dynamical steady states. The use of an idealized model is essential for these derivations. These points have been emphasized in Secs. I and VI of the main text.

Re: fluctuations: This comment is somewhat technical. It refers to the characterization of large-scale behavior for (less idealized) models where exact hydrodynamic limits are not available. Fluctuation effects are most commonly discussed in the context of critical points and the associated scaling exponents, as analyzed via the renormalization group (RG). In the current context, this means that the APLG has the exponents of the Gaussian RG fixed point, while one would expect “standard” (less idealized) models to have different exponents. This issue is discussed briefly in Secs. A.1 and D.2 of the SI. However, none of the conclusions of this manuscript rely on the critical behavior, so we have chosen not to discuss this point in detail. It is also possible that fluctuations strongly affect the behavior away from critical points, as happens (for example) in bubbly phase separation for active systems. This would be very interesting – even if this is the case, it is still useful to a reference system like the APLG, where fluctuations are suppressed. The possibility of bubbly phases is also now mentioned in Sec. A.1 of the SI.

2. *I find section IV to be rather weaker than the rest of the paper. For someone not familiar with the “multi-scale matched asymptotic method”, it feels like a bit of “cooking” to extract a solution with uncontrolled approximations. I am ready to be convinced that the extracted*

solution is close to the real one but this is not shown in the paper. I believe the authors should at least compare the solutions obtained with this method and the direct integration, for example with the traveling states seen in Fig. 7b.

The method of matched asymptotic expansions is a well-established systematic method to solve PDE problems involving small parameters and boundary layers. We already had a reference to a textbook on Perturbation methods by Holmes [48] with a chapter dedicated to this method. We have emphasized in the main text and in the Methods section that this is a systematic method that results in a controlled approximation in terms of the small parameter $\epsilon = 1/L$. The bottom row of Fig. 5 includes a comparison between the solution of the original problem (Eq. (7) in the moving frame) and the matched asymptotics leading-order problem (8).

Our revised manuscript includes two further comparisons between the asymptotic solution and the original one. First, we have added a new figure to the main text (Fig. 6), showing the convergence of the full solution to the problem Eq. (7) (corresponding to finite L) to the asymptotic solution to the problem Eq. (8) as we increase the value of L from 10 to 100. As expected, as L increases, the interface becomes sharper, the magnetisation away from the interface reduces, and the solution agrees better with the asymptotic prediction.

Second, in Subsection D.3, we include the spinodal calculation for the asymptotic solution. We compare the full calculation (which involves studying the eigenvalues of a 3×3 matrix in terms of the wavenumber q) and the reduced calculation (which considers the $q \ll 1$ limit and reduces to the study of a 2×2 matrix that we show that agrees with the outer problem (8)). The resulting two spinodals are indistinguishable, supporting the validity of our approach.

3. *One issue that I have is that in similar situations when one is looking for nonlinear solutions (e.g. FKPP fronts or, perhaps closer to the present topic, the patterns seen in flocking models as in Solon et al, Phys. Rev. E 92, 062111 (2015)), there exists, for the same parameters, a family of patterns with different velocities. Here, the authors suggest that the velocity is selected. Can they comment on why and how?*

We have given a particular initialization that provides a particular solution with a given speed. As shown in Fig. 7, different initializations may lead to different speeds. We expect that if we initialized the system with two wavelengths in the domain $[-L/2, L/2]$, then we would get a different speed. There are also zero-velocity solutions (PS solutions).

We have added a sentence about multiple speeds in the multistability section (Sec. V) and included the suggested reference in the discussion in Sec. VI.

4. *I tried to figure out how many particles there are in the simulations of Fig. 2 but could not based on available parameters (one cannot infer ℓ_x only from Pe). Please indicate all the parameters.*

The first paragraph of Sec. III states “The total number of particles is approximately $(\ell_y/\ell_x)\phi(L/h)^2$ ”, and the values of all these things are given in either the caption or in Methods. (We provide the number of particles as “approximate” because the initialization scheme decides separately whether to add a particle on each site, so there are fluctuations in the total number of particles.)

5. *p6: what are “natural” solutions of (7)?*

We meant smooth solutions. This has been amended in the text.

6. p6: Citing Ref. 44 here is rather strange as it comes after many papers on traveling bands in flocking models: the first bands in the Vicsek model are seen in Chaté et al, *Phys. Rev. E* 77, 046113 (2008), the first traveling phase separated state in the Active Ising model in Solon and Tailleur, *PRL* 111, 078101 (2013).

The formulation “traveling phase-separated states (traveling bands)” is also unfortunate because “bands” (an extensive number with a finite size, as in the Vicsek model) are usually distinguished from phase-separated states (of size proportional to system size, as in the active Ising model).

We have removed the reference to traveling bands (which is anyway not directly relevant), and we replaced the reference with the appropriate reference to Solon and Tailleur (Ref. [47])

Referees #3 and #4

The present work considers a mixture of passive and active particles in 2 spatial dimensions (2D). The authors first provide a microscopic agent based model (Active Passive Lattice Gas Model), where the passive particles diffuse while the active particles self-propel (in the x-direction only to allow for a later thorough analytic treatment) according to a discrete poisson variable (left/right). Passive (resp. active) particles remain passive (active) during all the simulation. The authors then build on previous results in the literature to derive an hydrodynamic continuous model. The results of the simulations for the microscopic and hydrodynamic models are compared: they are qualitatively and quantitatively similar.

The authors investigate the phase diagram in terms of the active/passive densities, showing that the binodal and the spinodal intersect. This opens the door to interesting dynamical phenomenology, here propagating waves, a ubiquitous feature in nonreciprocal active matter. In particular, they describe the dynamics of two similar phenomena: Counter-propagating (CP) and Traveling (T). T waves are isolated density fronts that propagate unidirectionally, without perturbation. CP waves, if we got it correctly, result from the interaction/collision of 2 T waves travelling in opposite directions. The authors used a multi-scale method of matched asymptotic expansions to obtain the profiles (in space and time) of travelling waves. The manuscript content is dense, complete and technical but remains understandable. Its main scope is to establish rigorous (generic) results based on a simple model that could eventually serve as a reference to understand the dynamical pattern formation behaviour of more complex active systems, where an analytical treatment is out of reach. I believe this objective is accomplished, although I wonder if Nature Communications is the right place for such fundamental, formal study, on a specific field within active matter physics. As one finds in the manuscript their “approach [...] complements the generic description of nonreciprocal systems via the NRCH equation”. Besides that, the manuscript should be improved before publication, to clarify the points we list below.

We thank the referees for their helpful comments. Regarding the suitability of these results for Nature Communications, we have added extra information in the paper to explain how our results for this idealized model are useful for the broad community of scientists studying nonequilibrium systems, including nonreciprocal pattern formation, active matter, etc. (see, in particular, Sec. VI.b).

Major comments:

1. *Major comment on the nonreciprocity: In addition to the title, the authors refer to their model system as being ‘nonreciprocal’ and the relation with nonreciprocity is mentioned several times throughout the main text:*

“Our exact hydrodynamic equation differs from NRCH, but the resulting phenomenology is similar and consistent with generic principles of self-organization via nonreciprocity.” (Mason et al., 2024, p. 1)

“This work addresses these challenges by analyzing a specific nonreciprocal system for which exact mathematical results can be derived” (Mason et al., 2024, p. 1)

“Our microscopic model clearly exhibits the principles of nonreciprocal self-organization.” (Mason et al., 2024, p. 2)

“In particular, (1) includes nonreciprocal couplings between active and passive densities.” (Mason et al., 2024, p. 3)

However, we don’t understand what the authors mean by ‘nonreciprocal’ in their system. We don’t see any source of nonreciprocity in the features of the dynamics written under Section I ACTIVE-PASSIVE LATTICE GAS (APLG) MODEL: (i) is pure diffusion, (ii) is self-propulsion, (iii) is excluded volume and (iv) is a Poisson process for the orientation of the active particles. The authors suggest (quite vaguely) that nonreciprocity arises from the bidispersity, mixing passive and active particles. We don’t understand why. On the other hand, neither we clearly identify the source of nonreciprocity in Eq. (1). Could the authors please detail this important point? maybe providing their definition of nonreciprocity and stating explicitly where does it appear in their micro and continuum model?

See response to comment 1 of referee 1.

2. *Major comment on the Counter Propagating waves, FIGURE 3:*

This figure is important to understand the phenomenon of interest: the 2 counter-propagating waves. However, it is not clear to us what happens precisely during the collision for the two clusters to leave with a higher speed/acceleration, before coming back to a constant speed (straight lines in the kymograph). Moreover, the authors write “The clusters slow down during collisions, but they eventually pass through each other.” It looks like the opposite to us, they accelerate; higher slopes in the $t - x$ plane means higher velocity. Or did we misunderstood something? Anyway, this point needs further clarification.

The referees are right that in the example of Fig. 3, the clusters accelerate rather than slow down during collisions. We have amended the text and added a reference to a simple example of this effect involving hard-core left- and right moving particles (no diffusion or tumbles) [RTB20], where one can quantify the acceleration exactly in terms of the transfer of “labels” due to the hard-core interaction (Fig. 6 in [RTB20]).

Also, as a side note on the choice of the colours of the first row: as the yellow usually is a background colour, we first thought that the cluster was in red. It is just a minor comment as the conveyed message remains clear, thanks to the other 2 rows and since the patterns of the high/low density regions are eventually the same due to the periodic boundary conditions.

We agree that this was a poor choice of colorbar; we have changed the color scales in Figures 2, 3 and 4.

Also, it is not clear to us why the CP propagating waves are “sinusoidal” (p. 5, left column). The profiles (Fig4c) seem more gaussian or lorentzian.

When discussing the CP solutions in Fig. 3, we say, “This perturbation grows via an instability that involves counterpropagating sinusoidal waves with equal speeds and growth rates.” We think there might be some confusion between the sinusoidal instability from the linearised problem and the resulting wave. We fully agree that the CP profiles are not sinusoidal in the dynamical steady state: this is because nonlinear terms become relevant after the initial exponential growth regime that comes from the instability.

(Note also that Fig. 4 shows a T profile. The relevance of this Fig for CP states relies on point 3 below.)

3. *More generally : The manuscript refers to the CP solutions and the T solutions as two distinct phenomenologies. How wrong is it to understand CP waves as a specific case of 2 single T waves ? Does the two waves constituting the CP solution have any additional properties compared to a single T wave ?*

CP solutions can indeed be viewed as two equal T solutions moving in opposite directions. However, it’s important to note that CP interfaces constantly alter their speed and height as their interaction with the approaching interface evolves over time.

We could only do the detailed analysis for T solutions. However, in numerical investigations, we found that the interfaces of CP solutions away from the collision times also followed the tie lines, just as T solutions do.

In the most general setting, solutions can comprise n interfaces. As interfaces tend to join together, the most common solutions observed are single (TP $n = 1$) or double (CP $n = 2$) interfaces. Depending on the position in the phase diagram, TP or CP solutions appeared to be the stronger local attractors in our numerical investigations.

Also, even though it seems complex, do the authors have an intuitive understanding of what happens for $t < 250$ in Fig.4(b) ? How does it compare to the CP solution ? Why does it collapses to a single T solution at one point, changing the symmetry of the pattern? Could this left/right spontaneous symmetry breaking also happen after a while in the true CP solution ?

The initial condition in the simulation of Fig. 4 is of the form $\rho_\sigma = \phi_\sigma + \delta\tilde{\rho}_\sigma$, where the perturbation $\tilde{\rho}_\sigma$ is a combination of left and right unstable sinusoidal models (obtained from the linear stability analysis) superimposed to a uniform random field (details are in Sec. E1 of the SI). Without the random perturbation, such an initial condition may have led to a periodic CP solution, but the random perturbation causes the symmetry breaking. This means the two counterpropagating waves are slightly different, and after four collisions, they coalesce into a single left-moving T solution. We have added a comment on this in the text.

Comparing Figure 3 and Figure S5. Figure 3 of the main text describes the CP solutions, composed of two waves. In this particular realisation, both waves are soft (smooth bumps in the density field). Figure S5 also describes a CP solution, the difference is that it seems to be a lot sharper, cf panel (a). An important scattering seems to occur in the kymographs of panel (b), could you explain why you don’t observe this in Figures 3 and 4 of the main text ?

The main difference between Figures 3 and S5 is that the system size in Fig. 3 is $L = 2$ whereas in Fig. S5 is $L = 100$. As predicted by our analysis in Sec. IV, the width of the interface scales with $1/L$ and hence we expect the interface in S5 to be much sharper indeed. We explain the lack of scattering in Figures 3 and 4 due to the much smaller domain, which restricts the types of patterns that can form.

Minor comments

1. *The active particle density ρ_a defined below eq. 1 is not used.*

This quantity is used in Eq. (11) and throughout the Supplementary Information. We have also now rewritten Eqs. (8) in terms of ρ_0 and ρ_a as it is more convenient for the discussion.

2. *The introduction of eq. 2 is grounded on the SEP: could the authors say a bit more on the origin of eq 2 in order to make the text somehow more self-contained?*

The quantity d_s (the self-diffusion constant of the SEP) appears in Eqs. (1)-(2) because it determines the rate at which individual particles diffuse through the system. (This is affected by the local density ρ .) The functional dependence is inherited from the SEP because the diffusive hops happen with a very large rate D_T/h^2 (as $h \rightarrow 0$) while other processes occur more slowly. Hence the dynamics on very short time scales (order h^2) resembles that of SEP. (Even so, particles are labelled by their orientations, so the relevant baseline is a three-species SEP, see for example [Qua92].)

3. *The black line in Fig 1 b is not mentioned in the caption neither explained. The grey region in this panel is also quite obscure.*

This has been clarified in the caption. (The grey region corresponds to total volume fractions bigger than 1, which are inaccessible due to size exclusion. The black tie line intersects the spinodal, which means that phase-separated states above this line are linearly unstable.)

4. *We wonder what would change if instead of having a passive-active mixture, one mixes particles with two different activity levels? Is there a qualitative difference? Is it the activity contrast the relevant parameter here, the density ratio, both? What is the role played by each one of this parameters? A discussion along these lines would help putting this results in the right context.*

This is a very interesting question. If instead of active and passive particles, we considered two types of active particles with volume fractions ϕ_1, ϕ_2 and activities or self-propulsion speeds $v_1 \neq v_2$, the hydrodynamic description would involve four densities $\rho_{1,\pm}, \rho_{2,\pm}$ instead of three. Provided that the symmetric jump rates (D_T) are equal, the hydrodynamic limit in this case is a straightforward generalization of Eq. (1):

$$\partial_t \rho_{i,\sigma} = \nabla \cdot [d_s(\rho) \nabla \rho_{i,\sigma} + \rho_{i,\sigma} \mathcal{D}(\rho) \nabla \rho] - \partial_x [\rho_{i,\sigma} s(\rho) M + \sigma \text{Pe}_i d_s(\rho) \rho_{i,\sigma}] - \sigma m_i, \quad (1)$$

where $i = 1, 2, \sigma = \pm$, $\text{Pe}_i = v_i / \sqrt{D_T D_R}$, $\rho = \rho_{1,+} + \rho_{1,-} + \rho_{2,+} + \rho_{2,-}$ is the total density, $m_i = \rho_{i,+} - \rho_{i,-}$ and $M = \text{Pe}_1 m_1 + \text{Pe}_2 m_2$. In (1), we have assumed that flips in direction happen at the same rate for both species.

This system of equations could be studied along similar lines to our analysis of the APLG, and we would expect similar phenomenology. However, this is beyond the scope of our paper, and we have not attempted it (so far).

Off-lattice active-active mixtures were studied computationally in [KK20] using active Brownian particles interacting with a WCA potential, finding both static phase-separated states as well as dynamical steady states. However, the referee's comments show that the parameter space is quite large, and it is not obvious which parameters are most important. Analysis of these effects is beyond the scope of this paper, but we have commented on the possibility of active-active mixtures in the discussion Sec. VI.

5. *It is mentioned that MIPS “is also a familiar phenomenon in nonreciprocal matter”. Here again a discussion is missing on what do the authors have in mind with nonreciprocity, as MIPS is typically referred to a phase transition emerging from the competition between self-propulsion and excluded volume, with no nonreciprocity (or at least not discussed with this jargon).*

We have edited this discussion, to clarify our point: Nonreciprocal systems like NRCH can exhibit static phase separation (PS) as well as dynamical pattern-forming states, two relevant examples are cited for this. In the cases where the nonreciprocal system is an active-passive mixture then the phase separation is indeed caused by a combination of self-propulsion and excluded volume (ie MIPS).

6. *PRECISION : “the mean-field approximation $\langle \eta_\sigma(\mathbf{x}, t) \eta_\sigma(\hat{\mathbf{x}}, t) \rangle \approx \rho_\sigma(\mathbf{x}, t) \rho_\sigma(\hat{\mathbf{x}}, t)$ does not hold and no explicit formula for $d_s(\rho)$ exists.” (Mason et al., 2024, p. 2) This technical point is interesting. Does it mean that there are very strong spatial correlations in the system or do I get it wrong ? Is it easy to provide an intuitive explanation?*

See point 4 in the reply to referee 1. At leading order the (local) distribution is mean-field-like. However, there are subleading corrections to this distribution that are strong enough to affect the hydrodynamic limit equation. (These are not “very strong spatial correlations” but they are also not negligible.) These points have been clarified in Sec. A of the supplementary information. (Some further intuitive arguments are given in the reply to point 4 of referee 1.)

7. *“[Inside the spinodal, the homogeneous state is linearly unstable, and the phase-separated (PS) state is stable; between the spinodal and binodal, the PS state is globally stable while the homogeneous state is metastable].” (Mason et al., 2024, p. 3) You can remove the brackets, this technical comment is useful to non-expert readers.*

We have removed the brackets, as suggested by the referees.

8. *FIGURE 4: It would help the reader to add an arrow indicating the direction of propagation of the T propagation. I understand why there is a peak in the passive density in front of the profile (the passive particles are effectively “pushed” to the left by the active ones: the active ones want to go left so they don’t move until the passive one diffuse to the left, and the passive particles can only diffuse to the left due to the combined effects of active particles being to their right + excluded volume. However, I don’t think I understand why the active density exhibits such a sharp profile, which explains the majority of the total peak in density of those T waves. Could the authors please explain this point ?*

We have added an arrow in Figure 4(a) to show the direction of propagation.

As the referees correctly point out, the passive particles are being pushed towards the left by the active particles, whose orientations are polarised to the left. Note that Fig. 4 shows a system of size $L = 25$ and the (“sharp”) width of the polarised region is of order 1. The reason is the same as for the (similarly sharp) domain walls in the phase-separated state: any region with polarisation m of order unity also has density gradients of order unity. We added a comment about this in Sec F of the SI.

9. *FIGURE 5 (now Figure 6): This good plot contains a lot of details. However, you might consider*

- *increasing the size of the points $P1 P2 P3 P4$, we don’t see them clearly even after a*

zoom, especially in the bottom row.

- indicating the densities (ϕ_a, ϕ_p) used for each column on top of each top row panel. You have the space and it would help the reader to know straight away what changes from one column to the other.
- changing the black cross symbol to another one, the black cross has been already used on the phase diagram (Fig. 1) to specify the bifurcation of co- dimension 2.

We have modified Figure 6 as suggested.

10. “Reducing ϕ_p ” (Mason et al., 2024, p. 7) : you indeed reduce ϕ_p but you mostly reduce ϕ_a , to exit the Dynamical region. . . .

We have adjusted the text to clarify how the parameters change.

11. SM Figure 5 : “Density profile at $t = 500$ ” (Mason et al., 2024, p. 19). In the title of panel (a) it’s written $t = 4000$.

We thank the referees for spotting this typo; we have now amended the caption to say $t = 4000$.

12. Typos “Parameters”, below Eq (S60) and in the last paragraph: with paramters $reltol=1e-8$, $abstol=1e-8$

We have corrected the typo.

References

- [Eri21] Clément Erignoux. Hydrodynamic limit for an active exclusion process. *Mémoires de la Société Mathématique de France*, 169:1–206, 2021.
- [KEBT18] Mourtaza Kourbane-Houssene, Clément Erignoux, Thierry Bodineau, and Julien Tailleur. Exact hydrodynamic description of active lattice gases. *Physical Review Letters*, 120(26):268003, June 2018.
- [KK20] Thomas Kolb and Daphne Klotsa. Active binary mixtures of fast and slow hard spheres. *Soft Matter*, 16(8):1967–1978, 2020.
- [MEJB23] James Mason, Clément Erignoux, Robert L. Jack, and Maria Bruna. Exact hydrodynamics and onset of phase separation for an active exclusion process. *Proc. R. Soc. A Math. Phys. Eng. Sci.*, 479(2279):20230524, November 2023.
- [MJB23] James Mason, Robert L. Jack, and Maria Bruna. Macroscopic behaviour in a two-species exclusion process via the method of matched asymptotics. *J. Stat. Phys.*, 190(3):47, January 2023.
- [Qua92] Jeremy Quastel. Diffusion of color in the simple exclusion process. *Commun. Pure Appl. Math.*, 45(6):623–679, 1992.
- [RTB20] Tertius Ralph, Stephen W Taylor, and Maria Bruna. One-dimensional model for chemotaxis with hard-core interactions. *Physical Review E*, 101(2):022419, 2020.
- [SWMC15] Joakim Stenhammar, Raphael Wittkowski, Davide Marenduzzo, and Michael E. Cates. Activity-induced phase separation and self-assembly in mixtures of active and passive particles. *Physical Review Letters*, 114(1):018301, January 2015.

[WWG16] Adam Wysocki, Roland G. Winkler, and Gerhard Gompper. Propagating interfaces in mixtures of active and passive Brownian particles. *New J. Phys.*, 18(12):123030, December 2016.

Title: Dynamical patterns and nonreciprocal effective interactions in an active-passive mixture through exact hydrodynamic analysis

Authors: J. Mason, R. L. Jack and M. Bruna,

We thank the referees for their careful reading of our revised manuscript. Below, we provide a summary of the main changes and a point-by-point response to the specific comments raised by reviewer #3. The equation references in the referee report and our response correspond to those in the revised manuscript. For convenience, the revisions to the manuscript are highlighted in red.

Reviewer #1

I have read the new manuscript and find it to be much improved. Specifically, the protrusion of the spinodal through the binodal is emphasized - a result that is likely to be found to be important in other systems. Other comments have also been clarified. I therefore recommend publication.

Reviewer #2

I am satisfied by the reply of the authors and thus recommend publication.

Reviewer #3

The authors have addressed the points raised in our previous report in a satisfactory way. I now understand better the scope of the work and its fit in Nature Communications. Regarding the ‘non-reciprocity’ arising in the model, I still have a few doubts that I believe should be clarified before publication. As the work is motivated from the viewpoint of non-reciprocal interactions, this concept has to be as clear and rigorous as possible.

1. *In the discussion SM sec B 1, one needs to assume that the friction matrix is independent of the system’s micro-state. But then it is said that $\Gamma_0 = \Gamma(0)$. It is confusing, because otherwise one would get also gradients of Γ in eq. S7. As mentioned after S16, the friction matrix generically depends on the configuration of the system/field.*

This is not the case; in (S6) we are not assuming Γ to be independent of \mathbf{x} . To get from (S6) to (S7), we linearise around $\mathbf{x} = \mathbf{0}$ and the leading-order term, using $\nabla E(\mathbf{0}) = 0$ is $\Gamma_0^{-1}H_0$ as written. The gradients of Γ would appear at the next order term only.

2. *Regarding the non-reciprocal case discussion, for which the linearised equations or the matrix, is no longer symmetric, is also somehow unclear. Newton’s third law establishes the equivalence between forces between bodies, not their gradients, as concluded by S12. B_{NR} would be the gradient of the friction and force, and should also be written explicitly. Might be also useful to provide a simple example as the predator-prey problem mentioned, or a mixture of active-passive particles. Do we need to go to a coarse-grained description to see non-reciprocity in active particle systems?*

We thank the referee for this remark. Indeed, Newton’s third law relates forces and not their gradients. In order to put that in the context of reciprocity, it is helpful to consider pairwise interactions (which can always be introduced as a linearisation). We have expanded the explanation of nonreciprocity in the context of ODE systems and included a example with three particles.

3. *The discussion about the PDE system is very useful indeed. But the 'mechanistic' particle-based one, is not. From this discussion I conclude that non-reciprocity arises from having a mixture of active and passive particles, at the continuum level. I guess this is more general, and any mixture of active particles with different activity would also be non-reciprocal in this sense (we don't need passive particles in the model, but just more than one species).*

This is correct; a system with two species is the minimal ingredient for non-reciprocity. Non-reciprocity should be expected by default, with the system being reciprocal only for particular parameter choices, or if reciprocity can be traced back to a fundamental underlying principle such as Newton's 3rd law.